EMBO
Molecular Medicine

# Inhibition of asparagine synthetase effectively retards polycystic kidney disease progression

Sara Clerici [1,12], Christine Podrini [1,10,12], Davide Stefanoni [1], Gianfranco Distefano[1], Laura Cassina [1], Maria Elena Steidl[1], Laura Tronci[2,3], Tamara Canu [4], Marco Chiaravalli[1], Daniel Spies[1,5], Thomas A Bell 3rd[6], Ana SH Costa[7,11], Antonio Esposito[4], Angelo D'Alessandro [8], Christian Frezza [9], Angela Bachi [3] & Alessandra Boletta [1✉]

## Abstract

**Polycystic kidney disease (PKD) is a genetic disorder characterized by bilateral cyst formation. We showed that PKD cells and kidneys display metabolic alterations, including the Warburg effect and glutaminolysis, sustained in vitro by the enzyme asparagine synthetase (ASNS). Here, we used antisense oligonucleotides (ASO) against *Asns* in orthologous and slowly progressive PKD murine models and show that treatment leads to a drastic reduction of total kidney volume (measured by MRI) and a prominent rescue of renal function in the mouse. Mechanistically, the upregulation of an ATF4–ASNS axis in PKD is driven by the amino acid response (AAR) branch of the integrated stress response (ISR). Metabolic profiling of PKD or control kidneys treated with *Asns*-ASO or *Scr*-ASO revealed major changes in the mutants, several of which are rescued by *Asns* silencing in vivo. Indeed, ASNS drives glutamine-dependent de novo pyrimidine synthesis and proliferation in cystic epithelia. Notably, while several metabolic pathways were completely corrected by *Asns*-ASO, glycolysis was only partially restored. Accordingly, combining the glycolytic inhibitor 2DG with *Asns*-ASO further improved efficacy. Our studies identify a new therapeutic target and novel metabolic vulnerabilities in PKD.**

**Keywords** ADPKD; Glutamine Metabolism; Metabolic Reprogramming; Antisense Oligonucleotides; Glycolysis
**Subject Category** Urogenital System

## Introduction

Autosomal dominant polycystic kidney disease (ADPKD) is one of the most common monogenic inheritable disorders. It is caused by loss of function mutations in either *PKD1* (83% of the cases) or *PKD2* (13% of the cases) genes, which encode for polycystins, PC1 and PC2, respectively, and a few minor genes (Torres et al, 2007; Harris and Torres, 2014; Ong and Harris, 2015). The inheritance of one mutated allele is not sufficient for the establishment of the pathology, thus it has been proposed that a second hit causing loss of heterozygosity is needed to drop polycystins' complex functional activity below a critical threshold (Qian et al, 1996; Watnick et al, 2000; Leeuwen et al, 2004; Hopp et al, 2012). The main clinical manifestation of the disease is bilateral formation and expansion of numerous cysts, that progressively compress and compromise the plasticity and function of the kidney. Secondary implications include cyst development in the liver and pancreas (Torres et al, 2007; Harris and Torres, 2014; Ong and Harris, 2015). In all these tissues, cysts are focal, clonal, and fluid-filled structures outpunching from the tubular epithelium (Qian et al, 1996). These cystic structures progressively increase in dimension and number during the lifespan of the patient (Harris and Torres, 2014; Bergmann et al, 2018). Furthermore, cardiovascular complications, hypertension, and aneurysm are important features of the disease (Bergmann et al, 2018). Notably, molecular mechanisms driving the cystogenic process are still not completely uncovered. However, different pathways are known to be dysregulated during the cyst expansion process, and are likely important to determine declining renal function.

Only one compound has been developed up to registration for this disease, based on an antagonist of the Vasopressin type II receptor (called Tolvaptan) that hampers cAMP production in the collecting ducts, an important driver of increased proliferation, as observed in preclinical models and in human samples (Torres et al, 2017). While the development of Tolvaptan proved the feasibility of

[1]Molecular Basis of Cystic Kidney Disorders Unit, Division of Genetics and Cell Biology, IRCCS, San Raffaele Scientific Institute, Milan, Italy. [2]Cogentech SRL Benefit Corporation, 20139 Milan, Italy. [3]IFOM ETS The AIRC Institute of Molecular Oncology, Milan, Italy. [4]Center for Experimental Imaging (CIS), IRCCS, San Raffaele Scientific Institute, Milan, Italy. [5]Center for Omics Sciences (COSR), IRCCS, San Raffaele Scientific Institute, Milan, Italy. [6]Ionis Pharmaceuticals, Carlsbad, CA, USA. [7]MRC, Cancer Unit Cambridge, University of Cambridge, Hutchison/MRC Research Centre, Box 197, Cambridge Biomedical Campus, Cambridge CB2 0XZ, UK. [8]Department of Biochemistry and Molecular Genetics, University of Colorado Denver, Aurora, CO, USA. [9]Faculty of Medicine and University Hospital Cologne, Faculty of Mathematics and Natural Sciences, Cluster of Excellence Cellular Stress Responses in Aging-associated Diseases (CECAD), Joseph-Stelzmann-Str. 26-50931, Cologne, Germany. [10]Present address: The BioArte Ltd, Laboratories at Malta Life Science Park (LS2.1.10, LS2.1.12-LS2.1.15), Triq San Giljan, San Gwann, SGN 3000, Malta. [11]Present address: Matterworks, Inc, 444 Somerville Avenue, Somerville, MA 02143, USA. [12]These authors contributed equally: Sara Clerici, Christine Podrini. ✉E-mail: boletta.alessandra@hsr.it

completing all the regulatory pathway for this complicated and life-long disorder, the molecule has limited efficacy, is poorly tolerated, and presents with quite important side effects such as rare, but severe liver toxicity (Torres et al, 2017; Watkins et al, 2015). Thus, the development of new therapeutic approaches for ADPKD is mandatory.

Our laboratory has contributed to defining ADPKD as a metabolic disorder, describing how the metabolic rewiring supports the required increase in proliferation observed during cystogenesis, and opening interesting opportunities for therapy (Rowe et al, 2013; Chiaravalli et al, 2016; Podrini et al, 2018, 2020). Metabolic reprogramming might be a particularly appealing dysfunction to be treated in the disease, as it is quite prominent and sustains the energetic requirements of proliferation. For instance, targeting the increased glycolysis (Warburg effect) using 2-deoxy-D-glucose (2-DG) treatment resulted in a prominent improvement of disease progression in preclinical animal models of the disease (Rowe et al, 2013; Chiaravalli et al, 2016; Riwanto et al, 2016; Nikonova et al, 2018; Lian et al, 2019). Abnormal cystic growth relies on the utilization of aerobic glycolysis, in a way resembling the Warburg effect observed in cancer, whereby cells lacking the polycystins become dependent on glucose for energy production (Rowe et al, 2013). Using metabolic tracing studies relying on heavy isotopologues, we have demonstrated that glutaminolysis is increased, in what we have described to be a compensatory mechanism able to fuel the TCA cycle (Podrini et al, 2018). Small amounts of this glutamine are utilized on the oxidative side of the TCA cycle allowing for the maintenance of mitochondrial membrane potential, even if glutamine utilization does not compensate for the severely reduced OXPHOS levels in $Pkd1^{-/-}$ cells (Menezes et al, 2016; Padovano et al, 2017; Podrini et al, 2018). Quite abundant levels of glutamine-derived α-KG are instead diverted towards reductive carboxylation, to generate citrate which provides abundant acetyl-CoA levels required to fuel fatty acids biosynthesis, sustaining membrane synthesis, and supporting proliferation, at least in vitro (Rowe et al, 2013; Menezes et al, 2016; Padovano et al, 2017; Podrini et al, 2018, 2020).

Thus, we and others have concomitantly highlighted cysts' dependence on glutamine utilization (Flowers et al, 2018; Podrini et al, 2018; Soomro et al, 2018), and we have further demonstrated that asparagine synthetase (ASNS) is a key player in the proficient utilization of glutamine to fuel TCA cycle and to cope with the increased energetic demand (Podrini et al, 2018). Indeed, silencing of Asns in vitro drastically impaired the glutamine utilization of $Pkd1^{-/-}$ cells leading to compromised survival and proliferation (Podrini et al, 2018). Of note, an increase in circulating asparagine has subsequently been reported in children and young adults with ADPKD (Baliga et al, 2021), supporting the likely increased expression or activity of this enzyme in PKD patients. Similarly, different types of solid tumors as well as leukemia have been associated with high expression levels of the enzyme ASNS, which can promote proliferation, metastasis, and chemoresistance (Chiu et al, 2020).

Here, we aimed to investigate the potential role of ASNS as a novel target for therapy acting on a metabolic vulnerability of ADPKD. We show that targeting Asns with antisense oligonucleotides (ASOs) in a slowly progressive orthologous model of PKD results in strong and remarkable disease improvement. Moreover, we identified general control nonderepressible-2 (GCN2) as a

major regulator of ASNS increased expression in this disease, mediated by an activation of the amino acid response (AAR) branch of the integrated stress response (ISR). Importantly, metabolomic profiling confirmed the increased asparagine levels in PKD kidneys and the concomitant upregulation of glycolysis and glutaminolysis. Notably, Asns-directed ASO rescue some of the metabolic alterations observed in PKD, without affecting glycolysis. In line with this, combining Asns inhibition with 2DG provides a further delay in disease progression. Finally, in the attempt to better understand the mechanism of action of ASNS upregulation in ADPKD, we found that its inhibition blunted the carbamoyl-phosphate synthetase 2, aspartate transcarbamoylase, and dihydroorotase (CAD)-dependent de novo pyrimidine synthesis pathway. Indeed, analysis of $^{13}C_5$-glutamine and $^{15}N_2$-glutamine tracing using cells lacking Pkd1 revealed an upregulation of this metabolic pathway.

Our data collectively demonstrate that targeting ASNS is a new and effective therapeutic approach that deserves to be further exploited.

# Results

## ASNS is upregulated in ADPKD

Asparagine synthetase (ASNS) is a transamidase that catalyzes the synthesis of asparagine (Asn) from aspartate (Asp) by deamidating glutamine (Gln) to form glutamate (Glu), through an ATP-dependent reaction (Lomelino et al, 2017). Despite being ubiquitous, ASNS is generally expressed at low levels in organs other than the exocrine pancreas (Milman and Cooney, 1974; Uhlén et al, 2015). However, the enzyme is tightly regulated as a biosensor of cell stress stimuli, and it was found to be induced in pathological conditions, including ADPKD (Lomelino et al, 2017; Podrini et al, 2018). Importantly, an increase in ASNS metabolic activity has been inferred in children and young adult ADPKD patients based on the increased levels of circulating asparagine (Baliga et al, 2021). Thus, we investigated whether ASNS was indeed upregulated using both in vitro and in vivo PKD models. Consistent with our previously reported transcriptional upregulation (Podrini et al, 2018), we found increased ASNS protein in mouse embryonic fibroblasts (MEF) knockout for the Pkd1 gene compared to controls, in basal conditions. Interestingly, ASNS expression was further increased in $Pkd1^{-/-}$ MEFs upon glucose deprivation both at the mRNA (Fig. 1A) and protein level (Fig. 1B), in line with the idea that ASNS mediates the glutamine compensatory utilization and with previous evidence in cancer (Chiu et al, 2020). Similar results were obtained using a novel renal epithelial cell line (murine cortical collecting duct cells, mCCD) invalidated for the Pkd1 gene by CRISPR/Cas9 (see "Methods", Figs. 1C and EV1A). Moreover, we observed increased ASNS protein expression in cystic $KspCre;Pkd1^{\Delta C/flox}$ kidneys, which develop early and severe PKD at postnatal day 4 (P4) (Fig. 1D) compared to relative controls. Further, analysis of published data from different PKD animal models, $Pkd1^{v/v}$ mice at P10 (Fig. 1E; Podrini et al, 2018) and $Pkd1^{RC/RC}$ mice at P12 (Fig. 1F; Olson et al, 2019) showed an increased Asns transcript in these models as well. We confirmed the upregulation of ASNS in ADPKD human samples microarray (Fig. 1G; Song et al, 2009), as previously

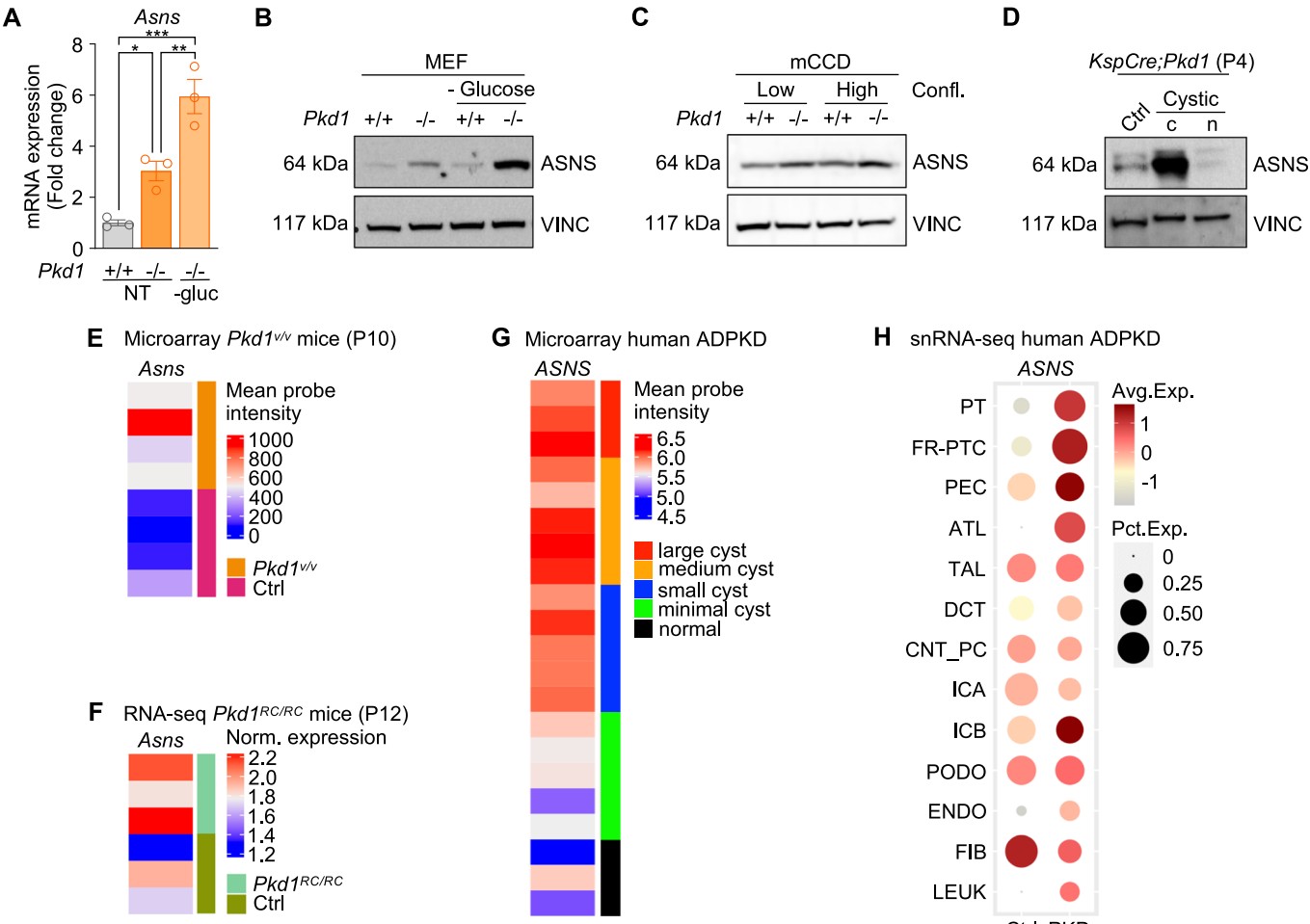

**Figure 1.  ASNS is upregulated in PKD models and in human ADPKD.**

(A) *Asns* mRNA expression in *Pkd1⁻/⁻* and control MEF, cultured in full medium or under glucose deprivation. Representative of *n* = 3 independent experiments. (B) ASNS expression in *Pkd1⁻/⁻* and control MEF, cultured in full medium or under glucose deprivation. Representative of *n* = 3 independent experiments. (C) ASNS expression in *Pkd1⁻/⁻* and control mCCD, cultured in 0% FBS medium under glucose deprivation for 24 h (low and high confluency). Representative of *n* = 3 independent experiments. (D) ASNS expression in *KspCre;Pkd1^{ΔC/flox}* (cystic) and relative control kidneys at P4. c cytoplasmatic fraction, n nuclear fraction. (E) *Asns* expression in P10 *Pkd1^{v/v}* mice microarray. (F) *Asns* expression in P12 *Pkd1^{RC/RC}* mice RNA-seq. (G) *ASNS* expression in ADPKD human samples microarray. (H) Dot plot of snRNA-seq dataset showing *ASNS* expression in clusters identified in human ADPKD cystic and normal kidney tissues. PT proximal tubule, FR-PTC failed-repair proximal tubular cells, PEC parietal epithelial cells, TAL thick ascending limb of Henle's loop, DCT distal convoluted tubule, CNT_PC connecting tubule and principal cells, ICA Type A intercalated cells, ICB Type B intercalated cells, PODO podocytes, ENDO endothelial cells, FIB fibroblasts, LEUK leukocytes. Data information: in (A) data are shown as mean ± SD. One-way ANOVA. **P* < 0.05; ***P* < 0.01; ****P* < 0.001. Source data are available online for this figure.

observed (Podrini et al, 2018). Finally, analysis of previously published datasets of single nucleus RNA-sequencing (snRNA-seq) of ADPKD patients (Muto et al, 2021) confirmed *ASNS* increase in clusters of proximal and distal tubules from cystic tissues compared to controls (Fig. 1H). Together, these data support the evidence that ASNS is strongly upregulated in multiple cellular and animal models as well as in human ADPKD samples.

## Targeting *Asns* ameliorates key features of PKD progression

We have previously demonstrated that ASNS is a central player in the metabolic rewiring occurring in PKD, being the preferential enzyme for glutamine usage in *Pkd1* mutant cells. In line with this,

we previously showed that the glutaminase enzyme (GLS) is not upregulated in *Pkd1* mutant cells and kidneys (Podrini et al, 2018). Indeed, the silencing of *Asns* was sufficient to impair proliferation and enhance apoptosis in *Pkd1⁻/⁻* cells (Podrini et al, 2018). Thus, we investigated the effect of targeting ASNS in vivo in an orthologous and slowly progressive PKD mouse model carrying a Tamoxifen-inducible Cre recombinase (B6.Cg-Tg(CAG-cre/Esr1*) 5Amc/J) enabling the inactivation at different time points (*Tam-Cre;Pkd1^{ΔC/flox}*) (Piontek et al, 2007; Wodarczyk et al, 2009). Given the lack of commercially available inhibitors of the enzyme, mice were treated with antisense oligonucleotides against *Asns* (*Asns*-ASO) or scramble controls (*Scr*-ASO). A pilot study was conducted to assess the efficacy of *Asns* silencing in a small cohort of mice carrying *Pkd1* gene inactivation upon Tamoxifen (250 mg/kg)

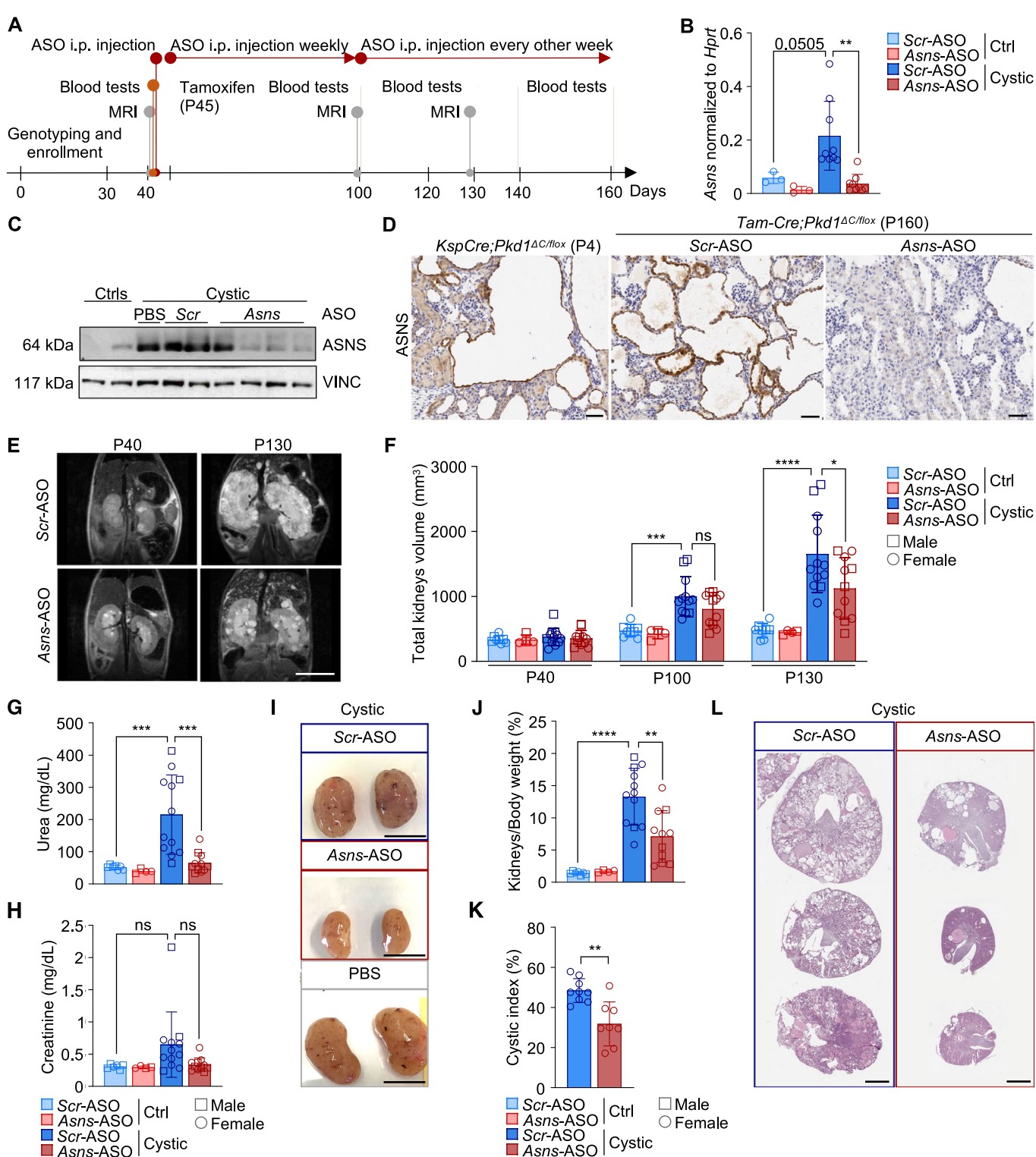

injection at P25 and developing a cystic phenotype within the next 70 days following a previously established protocol (Chiaravalli et al, 2016) (Fig. EV2A). Intraperitoneal (i.p.) injection of ASOs (50 mg/kg) weekly for 5 weeks and every other week untill the end of the study was sufficient to substantially reduce the *Asns* transcript in kidney tissues (Fig. EV2B), indicating that this dosage should be sufficient to silence the transcript. Of interest, we observed a trend of reduction in kidney volume (Fig. EV2C), kidney over body weight (Fig. EV2D), and blood urea nitrogen levels (BUN, Fig. EV2E), despite the low numerosity of this pilot cohort of mice. Histological analysis of transversal kidney sections showed a reduction in the percentage of cystic area in the

◀

**Figure 2. Targeting ASNS ameliorates PKD phenotype.**

(A) Experimental design of *Asns*-ASO-treatment study in *Tam-Cre;Pkd1*$^{\Delta C/flox}$ model (*n* = 8 (4M-4F) ctrls *Scr*-ASO; *n* = 4 (2M-2F) ctrls *Asns*-ASO; *n* = 12 (3M-9F) cystic *Scr*-ASO; *n* = 11 (5M-6F) cystic *Asns*-ASO). (B) *Asns* mRNA expression in cystic kidneys compared to controls at P160, treated with either *Scr*-ASO or *Asns*-ASO (*n* = 3 ctrls *Scr*-ASO; *n* = 3 ctrls *Asns*-ASO; *n* = 9 cystic *Scr*-ASO; *n* = 9 cystic *Asns*-ASO). (C) ASNS protein expression in cystic kidneys, treated with either PBS, *Scr*-ASO or *Asns*-ASO, compared to controls. (D) Representative images of IHC ASNS staining in cystic renal epithelium of *KspCre;Pkd1*$^{\Delta C/flox}$ (P4), and *Tam-Cre;Pkd1*$^{\Delta C/flox}$ (P160) mice treated with *Scr*-ASO or *Asns*-ASO. Scale bar (50 μm). (E) MRI representative images of cystic kidneys treated with *Scr*-ASO, *Asns*-ASO. Images were acquired at P40 (before tamoxifen induction) and at P130. Scale bar (1 cm). (F) Total kidneys volume of cystic and control mice treated with *Scr*-ASO or *Asns*-ASO, calculated at P40, P100 and P130 (*n* = 8 ctrls *Scr*-ASO; *n* = 4 ctrls *Asns*-ASO; *n* = 12 cystic *Scr*-ASO; *n* = 11 cystic *Asns*-ASO). (G) BUN concentration in cystic and control mice treated with *Scr*-ASO or *Asns*-ASO, measured at P160 (*n* = 7 ctrls *Scr*-ASO; *n* = 4 ctrls *Asns*-ASO; *n* = 12 cystic *Scr*-ASO; *n* = 11 cystic *Asns*-ASO). (H) Creatinine concentration in cystic and control mice treated with *Scr*-ASO or *Asns*-ASO, measured at P160 (*n* = 6 ctrls *Scr*-ASO; *n* = 4 ctrls *Asns*-ASO; *n* = 12 cystic *Scr*-ASO; *n* = 11 cystic *Asns*-ASO). (I) Representative images of cystic kidneys harvested at P160 from mice treated with *Scr*-ASO, *Asns*-ASO or PBS. Scale bar (1 cm). (J) Percentage of kidney weight normalized on total body weight at P160 of mice treated with *Scr*-ASO or *Asns*-ASO (*n* = 7 ctrls *Scr*-ASO; *n* = 4 ctrls *Asns*-ASO; *n* = 12 cystic *Scr*-ASO; *n* = 11 cystic *Asns*-ASO). (K) Quantification of the cystic area percentage of the total kidney area measured in sections of cystic *Scr*-ASO or *Asns*-ASO groups (*n* = 9 cystic *Scr*-ASO; *n* = 8 cystic *Asns*-ASO).
(L) Representative images of H&E stained cross sections of P160 cystic kidneys, treated with *Scr*-ASO or *Asns*-ASO. Scale bar (2 mm). Data information: in (B, F–H, J), data are shown as mean ± SD. One-way ANOVA. ns not significant; *$P < 0.05$; **$P < 0.01$; ***$P < 0.001$; ****$P < 0.0001$. In (K) data are shown as mean ± SD. Student's unpaired two-tailed *t* test. **$P < 0.01$. Source data are available online for this figure.

*Asns*-ASO-treated group, compared to *Scr* one (Fig. EV2F). Importantly, immunohistochemistry (IHC) directed against the *Asns*-ASO in the cystic kidneys from three independent litters revealed that these are extensively distributed in the cystic tissue, including staining of the epithelium lining the cysts (Fig. EV2G). Given the high variability of the model and the relatively low number of samples in this pilot study, the data pointed to a possible quite robust effect of improvement in disease progression upon silencing of *Asns*, and we set out to test the hypothesis using a more robust long-term experimental design.

We designed a study that could rely on a longer treatment using the same mouse model (*Tam-Cre;Pkd1*$^{\Delta C/flox}$), in which the *Pkd1* gene was inactivated by Tamoxifen (250 mg/kg) injection at postnatal day 45 (P45) leading to the development of a slowly progressive disease that enabled treatment for over four months (up to P160, Leeuwen et al, 2007). *Asns*- and *Scr*-ASOs (50 mg/kg) were i.p. injected weekly up to P100, and every other week thereafter until the end of the study. This approach allowed to analyze over time different parameters linked to pathology progression as indicated in the experimental design (Fig. 2A). At sacrifice, we confirmed that ASNS is induced in cystic kidneys at P160 and almost completely restored by *Asns*-ASO administration as compared to matching *Scr*-ASO-treated animals (Fig. 2B). Similar results were obtained when looking at the protein level, though some higher variability could be appreciated (Fig. 2C). To determine if the ASNS upregulation is occurring in the cystic epithelia, we first set out to test and validate a panel of anti-ASNS antibodies for their specificity by silencing the enzyme in cell cultures (Fig. EV1B,C). We did identify one antibody able to recognize a signal in *Pkd1*$^{-/-}$ cells that was practically abrogated by silencing the *Asns* gene. We used this antibody to stain the ASNS protein both in *KspCre;Pkd1*$^{\Delta C/flox}$ kidneys at P4, and *Tam-Cre;Pkd1*$^{\Delta C/flox}$ at P160 by IHC. We found a strongly positive signal in the cystic epithelia which was not detectable in the surrounding parenchyma. Importantly, in the kidneys treated with *Asns*-ASO the staining was drastically reduced, further validating the specificity of the antibody for IHC in vivo and demonstrating the effectiveness of the *Asns*-ASO in silencing the *Asns* gene specifically in the cystic epithelia (Figs. 2D and EV1D).

Magnetic resonance imaging (MRI) was applied to follow the progressive increase in kidney volume of the cystic mice after enrollment in the study. Interestingly, MRI scans along time

(Fig. 2E) and the relative quantification (Fig. 2F) highlighted a strong reduction in kidney volume at P130 in the *Asns*-ASO-treated group compared to the *Scr*-ASO one. Notably, *Asns*-ASO treatment completely rescued the renal function impairment, evaluated as blood urea nitrogen (BUN) (Fig. 2G), and partially rescued the increased creatinine levels (Fig. 2H). At the endpoint, *Asns*-ASO-treated animals showed reduced kidneys/body weight (Fig. 2J) compared to cystic *Scr*-treated ones, as clearly appreciable through representative images of an average experiment (Fig. 2I). Consistently, the histological analysis of kidney sections highlighted a reduction in the percentage of cystic area (Fig. 2K) in *Asns*-ASO group compared to *Scr* one, with some of the tissues analyzed showing an almost complete restoration of healthy tubules and parenchyma (Fig. 2L; Appendix Fig. S1). This study suggests that silencing *Asns* is a valuable approach to ameliorating key features of PKD and delaying disease progression likely acting directly on the cystic epithelia.

## The GCN2-dependent branch of the integrated stress response drives ASNS upregulation in PKD

Given the primary role of ASNS in supporting ADPKD progression, we investigated the mechanism that underlies its upregulation. ASNS expression and activity in humans are highly prompted by cell stress. Indeed, this enzyme is transcriptionally regulated by activating transcription factor 4 (ATF4), which sits at the crossroads of two main branches of the integrated stress response (ISR) (Fig. 3A), namely the amino acids response (AAR) and the unfolded protein response (UPR). Notably, we found that ATF4 is upregulated in all PKD models described above, including *KspCre;Pkd1*$^{\Delta C/flox}$ kidneys at P4 (Fig. 3B) and P45-tamoxifen-induced *Tam-Cre;Pkd1*$^{\Delta C/flox}$ kidneys at P160 (Fig. 3C). Of interest, the ATF4–ASNS axis is known to be activated in response to ER stress by PERK-dependent activation of unfolded protein response (UPR), a pathway that has been shown not to be deregulated in PKD (Fedeles et al, 2015; Roy et al, 2023), but also during an imbalance in amino acids availability in cells and tissues via activation of the General Control Nonderepressible-2-kinase (GCN2) within the so called amino acid response (AAR) that converges downstream with the PERK pathway on eIF2alpha which ultimately regulates the translation of ATF4 (Lomelino et al, 2017) (Fig. 3A). This process is mimicked by the limitation of essential

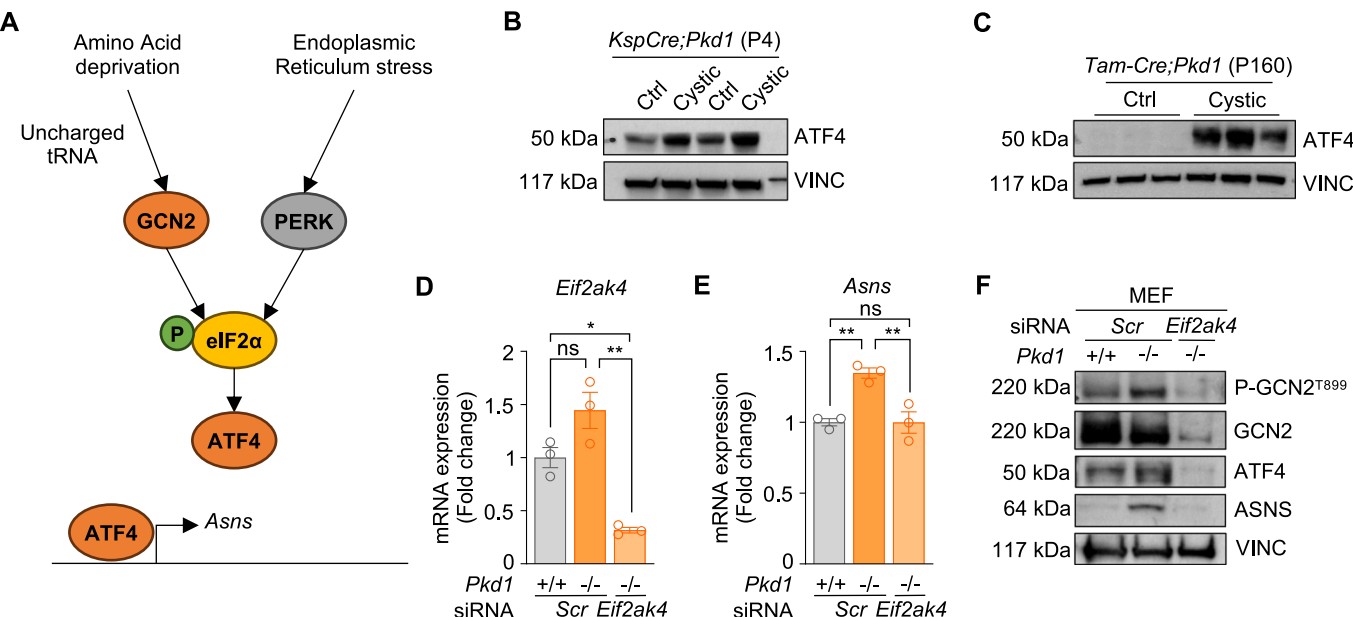

**Figure 3. ASNS is upregulated via GCN2-dependent activation of ATF4.**

(A) Schematic representation of the integrated stress response-dependent transcription of *Asns*. (B) ATF4 protein expression in *KspCre;Pkd1*$^{\Delta C/flox}$ cystic kidneys and relative controls at P4. (C) ATF4 protein expression in *Scr*-ASO-treated *Tam-Cre;Pkd1*$^{\Delta C/flox}$ mice and relative controls at P160. (D) *Eif2ak4* expression in *Pkd1*$^{-/-}$ and control MEF cells upon silencing (representative out of three independent experiments; $n = 3$ biological replicates). (E) *Asns* expression in *Pkd1*$^{-/-}$ and control MEF cells upon *Eif2ak4* silencing (representative of three independent experiments; $n = 3$ biological replicates). (F) P-GCN2, ATF4 and ASNS protein expression in *Pkd1*$^{-/-}$ and control MEF ± silencing of *Eif2ak4* ($n = 2$). Data information: in (D, E) data are shown as mean ± SD. One-way ANOVA. ns not significant; *$P < 0.05$; **$P < 0.01$. Source data are available online for this figure.

amino acids (EAAs) in cell culture (Jin et al, 2021), whereby GCN2 acts as a metabolic sensor for the depletion of amino acids, via the binding of uncharged tRNAs (Kanno et al, 2020). Our previous metabolomic analysis of *KspCre;Pkd1*$^{\Delta C/flox}$ cystic kidneys in newborn mice highlighted an impairment in the aminoacyl-tRNA pathway as well as an imbalance in amino acid biosynthesis (Podrini et al, 2018), suggesting that the amino acid response (AAR) could be the driver for ASNS upregulation in PKD. Thus, we investigated if ASNS expression is regulated by the GCN2-dependent branch of the integrated stress response (ISR). Indeed, we found a trend of upregulation of GCN2 transcript in *Pkd1*$^{-/-}$ MEF cells (Fig. 3D) along with an upregulation of the phosphorylation (Fig. 3F) levels, which was paralleled by ATF4 and ASNS increase. Furthermore, *Eif2ak4* (the GCN2 gene) silencing completely abrogated ATF4 upregulation and ASNS induction (Fig. 3E,F), indicating that this kinase is essential in the process. Collectively, these data show that *Pkd1* loss leads to upregulation of the amino acid response (AAR) which ultimately is responsible for the upregulation of the enzyme ASNS.

## *Asns*-ASO treatment rescues several metabolic pathways in PKD

In our previous studies, we identified ASNS as a key player in glutaminolysis in PKD. Therefore, we set out to determine whether *Asns*-ASO treatment impacts on PKD progression in our in vivo model by affecting key metabolic processes deregulated in the disease. To assess this, we performed targeted metabolomics by liquid chromatography–mass spectrometry (LC–MS) on cystic and control kidneys treated with *Asns*-ASO versus *Scr*-ASO. To this end, and to facilitate the identification of profound changes in metabolism, we selected the five cystic *Scr*-ASO-treated mice with the most exacerbated PKD phenotype, and the five cystic *Asns*-ASO-treated animals that presented the most striking improvement, as assessed by MRI scans at P130 (Fig. 4A). Profiling using LC–MS identified 162 metabolites, out of a total of 265, that were changing significantly ($P < 0.05$) between conditions. Notably, principal component analysis (PCA) revealed a clear clustering of controls *Scr*-ASO and *Asns*-ASO indicating a minimal impact of this treatment on WT kidneys, in line with the low levels of expression of this enzyme in basal conditions. Of note, *Scr*-ASO cystic kidneys clustered together and separated quite robustly from the controls. On the other hand, cystic kidneys treated with *Asns*-ASO clustered together and almost completely overlapped with the controls, indicating that the treatment rescued the metabolic derangement in PKD kidneys almost to completion, but had minimal effects on controls (Fig. 4B). The dendrogram (Fig. EV3A) and hierarchical clustering analysis (HCA) further confirmed this clustering of the samples (Fig. EV3B). Furthermore, heatmap representation clearly showed that targeting ASNS affects the metabolic processes altered in PKD cystic kidneys by rescuing 53.70% of the 162 metabolites altered in *Scr*-ASO ones (Fig. EV3B). We first confirmed that silencing of *Asns* significantly reduced asparagine levels, which were increased in cystic *Scr*-ASO kidneys, in line with the functional role of this enzyme. The substrates (aspartate and -glutamine), as well as the second product of the reaction (glutamate), were consistently changing in *Scr*-ASO versus *Asns*-ASO (Fig. 4C). Moreover, *Asns*-ASO treatment reduced

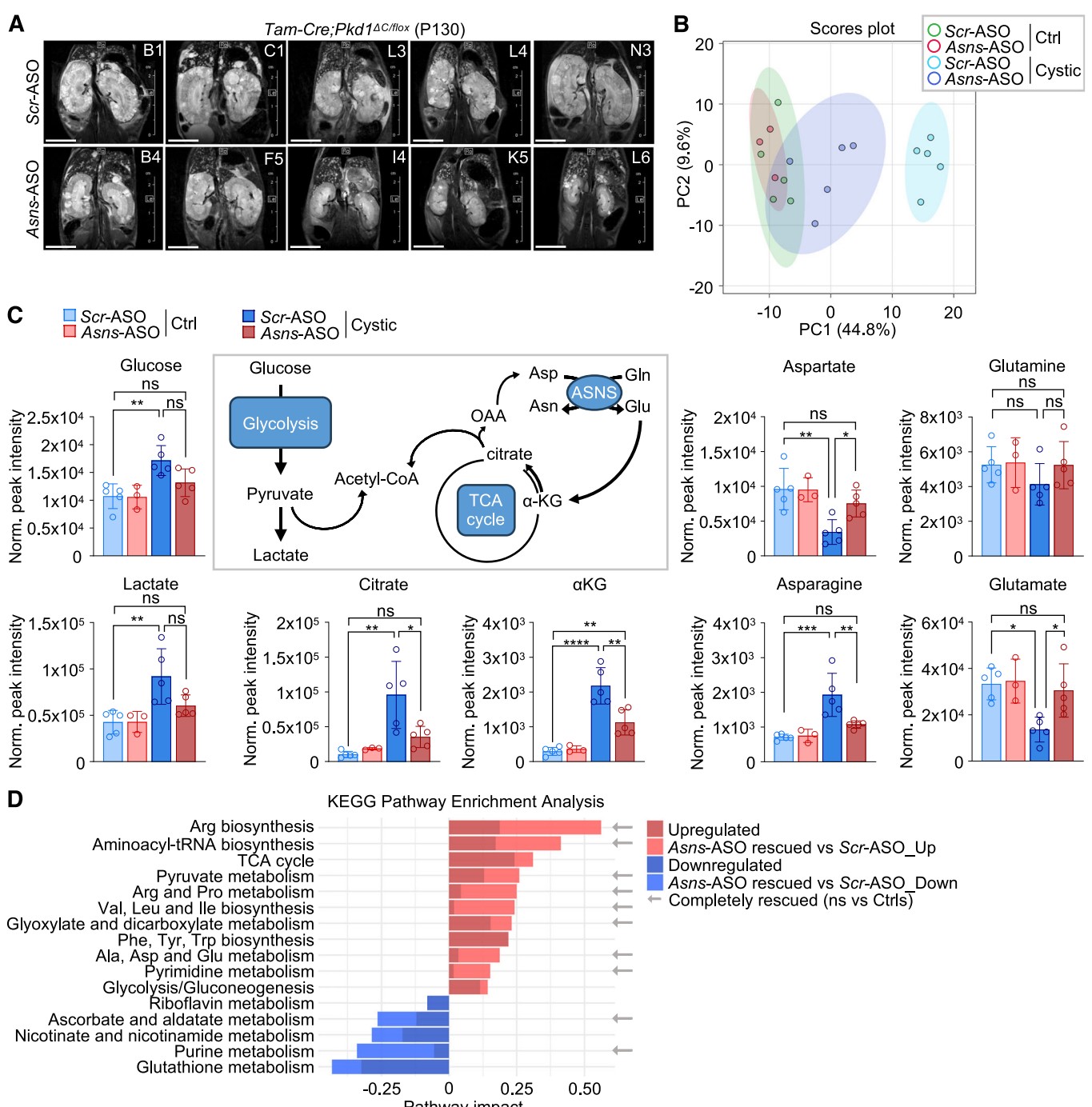

**Figure 4. *Asns*-ASO treatment rescues the metabolic reprogramming occurring in PKD.**

(A) MRI scan of P130 cystic *Tam-Cre;Pkd1^{ΔC/flox}* kidneys, treated with ASOs, included in the metabolomic analysis. Scale bar (1 cm). (B) Principal Component Analysis (PCA) of targeted metabolomics of *Tam-Cre;Pkd1^{ΔC/flox}* cystic kidneys and relative controls, treated with *Scr-* or *Asns*-ASO (n = 5 ctrl *Scr*-ASO; n = 3 ctrl *Asns*-ASO; n = 5 cystic *Scr*-ASO; n = 5 cystic *Asns*-ASO). (C) Schematic representation and analysis of metabolites involved in ASNS-dependent fueling of TCA cycle (reductive carboxylation) and glycolysis (n = 5 ctrls *Scr*-ASO; n = 3 ctrls *Asns*-ASO; n = 5 cystic *Scr*-ASO; n = 5 cystic *Asns*-ASO). (D) Pathway Enrichment Analysis of up- and downregulated pathways (Cystic *Scr*-ASO respect to the other groups), and rescue of *Asns*-ASO treatment compared to *Scr*-ASO group. Pathways non-statistically (ns) different from Ctrl group upon *Asns*-ASO treatment are indicated as completely rescued. Data information: In (C) data are shown as mean ± SD. One-way ANOVA, corrected with Tukey's multiple comparisons. ns not significant; *$P < 0.05$; **$P < 0.01$; ***$P < 0.001$; ****$P < 0.0001$. In (D) statistical significance of Pathway Enrichment and Impact was evaluated with Hypergeometric Test with Relative-betweenness Centrality, based on KEGG Database with $P < 0.05$, FDR $< 0.1$. Source data are available online for this figure.

Citrate and α-KG levels, which could be explained by a decrease in reductive carboxylation, in line with our previous observation in $Pkd1^{-/-}$ MEF cells upon silencing of *Asns* (Podrini et al, 2018), and resulting in glutamate accumulation (Fig. 4C). According to previous studies (Rowe et al, 2013), glycolysis was upregulated in cystic kidneys, in line with the robust Warburg effect observed in this disease, but *Asns*-ASO only partially reduced glucose and lactate levels (Fig. 4C). Indeed, pathway enrichment analysis based on KEGG database identified 11 upregulated (dark red) and 5 downregulated (dark blue) metabolic pathways significantly changed in cystic *Scr*-ASO kidneys compared to the other groups. Notably, 10 out of the 16 pathways were completely rescued by *Asns*-ASO treatment, as they were no longer significantly different compared to control groups (Fig. 4D). Interestingly, various pathways were related to amino acid metabolism and aminoacyl-tRNA biosynthesis (Fig. 4D) and rescued by the silencing of *Asns*. Indeed, we observed an imbalance of amino acids in cystic kidneys, involving some essential aa, which was corrected by *Asns*-ASO treatment (Fig. EV3C). This possibly supports the observation that *Asns* upregulation is mediated by the AAR branch of the ISR in cystic kidneys, as described in Fig. 3. Altogether, these results corroborate the central role of ASNS in the metabolic rewiring occurring in PKD, highlighting the therapeutic potential of its inhibition.

## ASNS sustains de novo pyrimidine synthesis in PKD

One of the central mechanisms through which ASNS sustains proliferation in various cancerous conditions is the increased de novo pyrimidine biosynthesis (Sullivan et al, 2015; Krall et al, 2016, 2021). In line with this, our data showed that this pathway was strongly upregulated in PKD kidneys, potentially contributing to support proliferation during the cystic process, and was completely rescued by *Asns* targeting (Fig. 4D). Indeed, looking at the individual metabolites identified in the LC–MS profiling of ASO-treated kidneys revealed that all three intermediate products (carbamoyl-phosphate, N-carbamoyl aspartate, dihydroorotate) of the multifunctional enzyme CAD (carbamoyl-phosphate synthetase 2, aspartate transcarbamoylase and dihydroorotase) were upregulated in cystic kidneys and rescued by *Asns*-ASO (Fig. 5A). Consistently, the intermediates of the following steps of nucleotide biosynthesis (orotate, orotidine-5'-phosphate) were increased in diseased kidneys and responsive to *Asns*-ASO treatment (Fig. 5A). Furthermore, the reduction of the substrates UMP (uridine-5'-monophosphate) and PRPP (5-phospho-α-D-ribose-1-diphosphate) supports the idea of a possible increased consumption for pyrimidine biosynthesis (thymine, cytosine, uracil), and it is also rescued by *Asns* targeting (Fig. 5A). Further to this, and in line with previous literature, our data also highlight a significant impact of the purine biosynthesis pathway as well, which however is reduced in PKD kidneys and it is rescued by *Asns*-ASOs (Fig. 5C) indicating that an imbalance between pyrimidine and purine levels might be observed in these tissues. Given the major effect of *Asns* targeting on pyrimidine biosynthesis, in line with prior literature supporting a role for this pathway downstream of ASNS in sustaining proliferation, and given the evidence that silencing of *Asns* impairs the proliferation of $Pkd1^{-/-}$ cells (Podrini et al, 2018), we tested whether *Asns*-ASO treatment could hamper cyst expansion in the $Tam\text{-}Cre;Pkd1^{\Delta C/flox}$ kidneys acting on the proliferative potential of

epithelial mutant cells. Thus, we quantified the percentage of Ki-67-positive nuclei in the epithelium lining the cysts of ASO-treated kidneys. We observed a reduction in the proliferation of *Asns*-ASO-treated kidneys of the pilot study at P94, compared to *Scr*-treated samples (Fig. 5B). We also attempted to measure the levels of apoptosis, but this was surprisingly extremely low in all conditions tested, including the *Asns*-ASO treatment. These data taken together suggest that *Asns*-ASO likely impacts cell proliferation, at least to some extent.

To further corroborate the metabolomic data highlighting the upregulation of the pyrimidine biosynthesis pathway in the long-term model of PKD, we investigated the expression of the first-step enzyme CAD in our animal models. In line with the increase in the products of the 3-step reaction of CAD (Fig. 5A), we observed an increase in the activatory phosphorylation S1859 of the enzyme CAD in PBS- and *Scr*-treated kidneys at P160, which is partially reduced by *Asns* silencing (Fig. 6A). Consistently, CAD was transcriptionally upregulated in different previously published datasets of PKD animal models, namely $Pkd1^{v/v}$ (Fig. 6B; Podrini et al, 2018) and $Pkd1^{RC/RC}$ murine kidneys (Fig. 6C; Olson et al, 2019), as well as in microarrays (Fig. 6D; Song et al, 2009) and in snRNA-seq (Fig. 6E; Muto et al, 2021) datasets from ADPKD patients, the last one mainly in the proximal tubule cluster of diseased kidneys. In addition, the analysis of previously published untargeted metabolomic data (Podrini et al, 2018) further confirmed an increase in the intermediates of de novo pyrimidine biosynthesis pathway downstream of CAD activity, in particular dihydroorotate, orotate, and orotidine in neonatal PKD kidneys from the $KspCre;Pkd1^{\Delta C/flox}$ mouse model at P4 (Fig. EV4A).

Interestingly, glutamine is a key carbon and nitrogen source for the de novo pyrimidine biosynthesis. Based on the results above, we, therefore, assessed whether the CAD-dependent metabolic intermediates derive from ASNS-driven glutamine utilization in $Pkd1^{-/-}$ cells. To this end, we used a $^{13}C_5$-glutamine tracing experiment followed by LC–MS on $Pkd1^{-/-}$ MEF cells with or without silencing of *Asns* by siRNA (Podrini et al, 2018). We found an increase in labeled N-carbamoyl aspartate (M + 4) in $Pkd1^{-/-}$ cells. Importantly, the silencing of *Asns* in these cells completely restored the metabolite levels down to baseline (Fig. 6F). The enzyme ASNS is a transamidase that produces asparagine and glutamate by transferring an amidic group from glutamine to aspartate. Therefore, we performed a new tracing study to follow the fate of the nitrogen groups by using $^{15}N_2$-glutamine. In this case, we could confirm the increased uptake of M + 2 glutamine and production of M + 1 glutamate and asparagine, in line with the increased catalytic activity of ASNS in $Pkd1^{-/-}$ MEF cell lines, all changes that were abrogated by silencing of *Asns* (Fig. 6G). Of interest, in these tracing studies we could confirm that the pyrimidine biosynthesis pathway is increased and rescued by silencing of *Asns*. In line with previous literature, we also observed that the increased phosphorylation of CAD was rescued by the mTORC1 inhibitor rapamycin (Fig. EV4B); however, *Asns* upregulation in these cells is not rescued by rapamycin (Fig. EV4B). These data collectively demonstrate that the mTORC1 cascade is participating into the pathway by driving the activation of the enzyme CAD. However, the contribution of ASNS catalytic activity to the pyrimidine biosynthesis pathway is independent of mTORC1. Thus, we conclude that the required role of ASNS for glutamine utilization does not only fuel the TCA cycle and

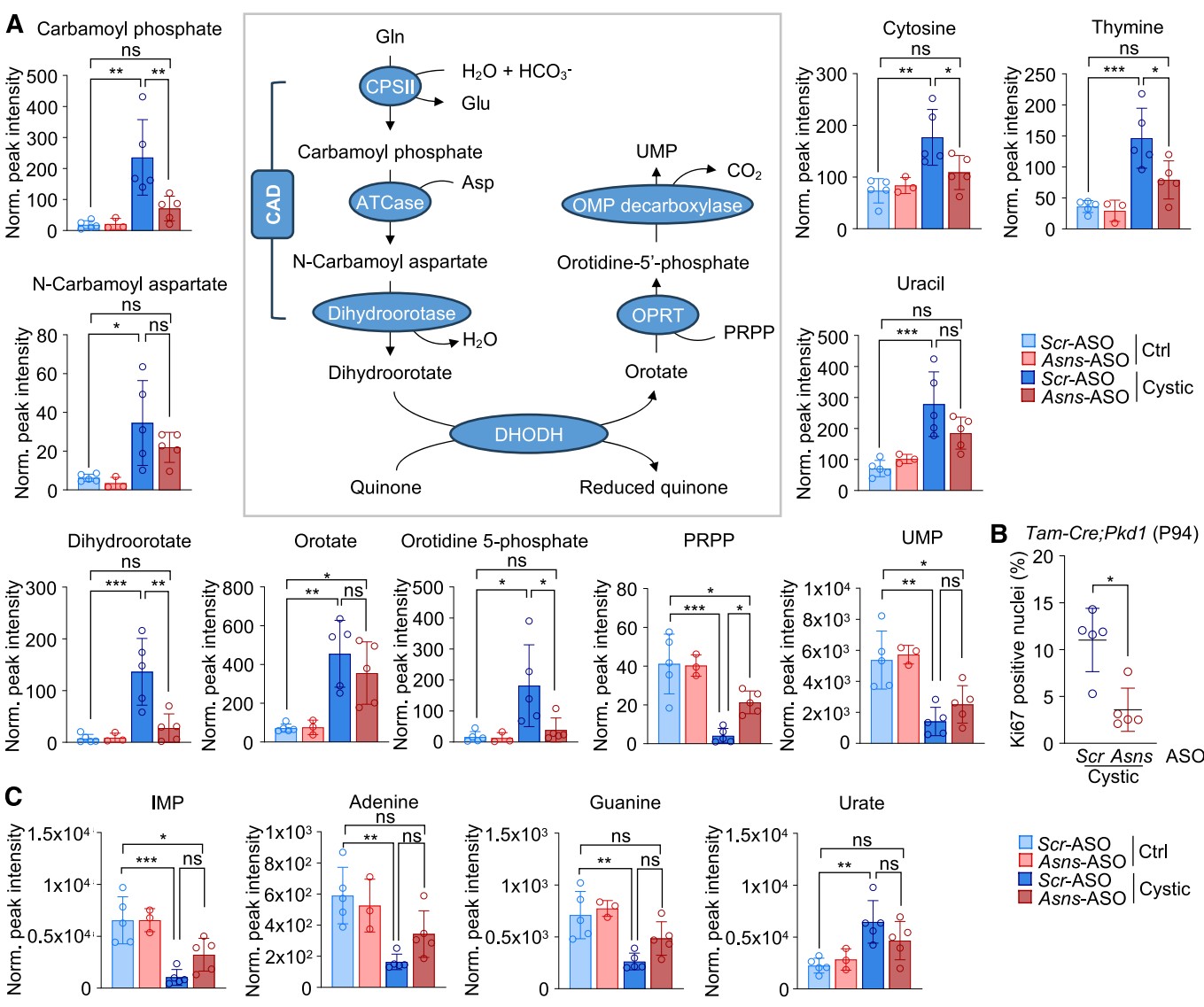

**Figure 5. Targeting ASNS hampers CAD-dependent pyrimidine biosynthesis in the PKD model.**

(A) Schematic representation CAD-dependent de novo pyrimidine biosynthesis pathway and targeted metabolomic profile of relative intermediates normalized on protein content, of P160 *Tam-Cre;Pkd1^{ΔC/flox}* cystic and control kidneys treated with ASOs (n = 5 ctrls *Scr*-ASO; n = 3 ctrls *Asns*-ASO; n = 5 cystic *Scr*-ASO; n = 5 cystic *Asns*-ASO). (B) Quantification of Ki-67-positive nuclei in the epithelium lining the cysts in the kidney cortex of *Tam-Cre;Pkd1^{ΔC/flox}* mice at P94, treated with *Scr*- or *Asns*-ASO (n = 5 cystic *Scr*-ASO; n = 5 cystic *Asns*-ASO). (C) Targeted metabolomic of main products of purine de novo biosynthesis pathway normalized on protein content, of P160 *Tam-Cre;Pkd1^{ΔC/flox}* cystic and control kidneys treated with ASOs (n = 5 ctrls *Scr*-ASO; n = 3 ctrls *Asns*-ASO; n = 5 cystic *Scr*-ASO; n = 5 cystic *Asns*-ASO). Data information: In (A, C) data are shown as mean ± SD. One-way ANOVA, corrected with Tukey's multiple comparisons. In (B) data are shown as mean ± SD. Student's unpaired two-tailed *t* test. ns not significant; *$P < 0.05$; **$P < 0.01$; ***$P < 0.001$. Source data are available online for this figure.

reductive carboxylation as we previously showed (Podrini et al, 2018), but it also enhances the pathway of de novo pyrimidine biosynthesis likely explaining the dual role of ASNS in driving proliferation and allowing survival in cells lacking the *Pkd1* gene that we previously described.

## Analysis of non-rescued pathways identifies an opportunity for combination therapy in PKD

ASNS inhibition had a great effect on the metabolic derangement of PKD and rescued several pathways, and some of them to

completion (i.e., the pathway was no longer significantly different from controls). However, we noticed that some other pathways were not as strongly affected (i.e., they were significantly improved over *Scr*-ASO, but still significantly different from controls). We thus set out to analyze the pathways for which we observed only a partial rescue by treatment with *Asns*-ASO, because in principle these should offer the opportunity to combine a second treatment to improve efficacy. Among these pathways, we found glutathione metabolism, nicotine, and nicotinate metabolism, and more pronouncedly glycolysis (Fig. 7A). We previously proposed, based on our in vitro studies that glutamine compensates for the reduced

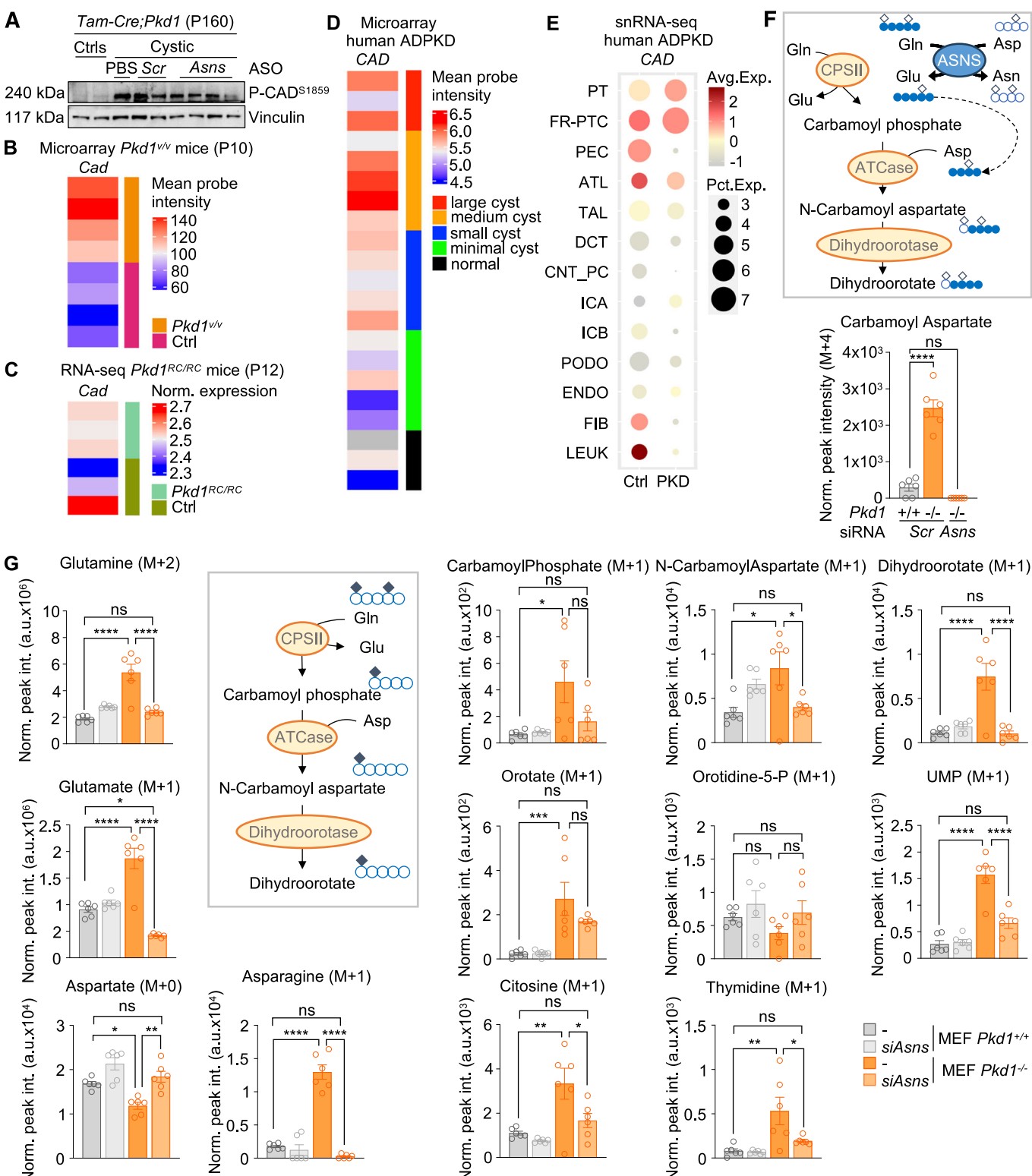

utilization of glucose in the TCA cycle, and therefore combining glutamine with glucose starvation, or exposing cells silenced of *Asns* to glucose starvation further enhances the effective retardation of cell growth (Podrini et al, 2018). Thus, given that the glycolytic pathway was only partially affected in PKD kidneys treated with

*Asns*-ASO prompted us to add a glycolytic inhibitor that we have previously characterized (2DG, Chiaravalli et al, 2016) to the *Asns*-ASO therapy. We initiated a new study designed as the previous one (Fig. 2), in which 2DG (2-deoxyglucose, the metabolically inactive analog of glucose) was added to the *Asns*-ASO or *Scr*-ASO

**Figure 6.   CAD-dependent pathway is upregulated in different models of PKD and rescued by *Asns* silencing in vitro.**

(A) Western blot analysis of *Tam-Cre;Pkd1^ΔC/flox* and relative controls at P160 treated with *Asns*-ASO or *Scr*-ASO. (B) *Cad* expression in microarray from cystic kidneys of P10 *Pkd1^fl/v* mice, compared to relative controls. (C) *Cad* expression in RNA-seq of *Pkd1^RC/RC* cystic mice at P12, compared to controls. (D) *CAD* expression in microarray of ADPKD patients samples. (E) Dot plot of snRNA-seq dataset showing *CAD* expression in clusters identified in human ADPKD cystic and normal kidney tissues. PT proximal tubule, FR-PTC failed-repair proximal tubular cells, PEC parietal epithelial cells, TAL thick ascending limb of Henle's loop, DCT distal convoluted tubule, CNT_PC connecting tubule and principal cells, ICA Type A intercalated cells, ICB Type B intercalated cells, PODO podocytes, ENDO endothelial cells, FIB fibroblasts, LEUK leukocytes. (F) Tracing metabolomics analysis ($^{13}C_5$-glutamine) evaluating labeled Carbamoyl Aspartate (M + 4) in *Pkd1^−/−* and control MEF cells ± silencing of *Asns* (n = 6 biological replicates in one experiment). (G) Tracing metabolomics analysis ($^{15}N_2$-glutamine) evaluating labeled metabolites of the de novo pyrimidine biosynthesis pathway in *Pkd1^-/-* and control MEF cells ± silencing of *Asns* (n = 6 biological replicates in one experiment). Data information: in (F, G) data are shown as mean ± SEM. One-way ANOVA. ns not significant; *$P < 0.05$; **$P < 0.01$; ***$P < 0.001$; ****$P < 0.0001$. Source data are available online for this figure.

treatment in the *Tam-Cre;Pkd1^ΔC/flox* mouse model where cystogenesis was induced at P45 by Tamoxifen injection. Mice were next treated daily with gavage 2DG (100 mg/kg) or PBS and with either *Asns*-ASO or *Scr*-ASO (50 mg/kg), administered i.p. weekly up to P100 and every second week up to P160 (Fig. 7B). Results showed that the combined treatment reached a similar improvement of disease at the endpoint in the *Asns*-ASO group as compared to *Scr*-ASO-treated mutants (P160, Fig. 7E,F). However, combined treatment enhanced efficacy at an earlier time point as compared to *Asns*-ASO alone, as evidenced by a significant reduction in kidney volume measured by MRI scan at P100 (Fig. 7C,D). Indeed, the side-by-side comparison between the two studies showed that all the animals enrolled in the combinatory study (COMBO) were consistently responsive to the treatment already at P100 in terms of kidney volume reduction, while in ASO study we observed a group of animals in which the response is appreciable only at later time points (Fig. 7G). Collectively, these data indicate that adding an inhibitor of glycolysis (2DG) to the ASNS silencing enhances the effectiveness of the therapeutic approach at early time points, opening an opportunity for combination therapy by exploiting the reduced metabolic flexibility in PKD.

## Discussion

In this study, we demonstrate that the metabolic enzyme ASNS is a valuable therapeutic target in PKD. We show that this enzyme is upregulated in cells and tissues derived from murine or human PKD kidneys, and that using antisense oligonucleotides (ASOs) directed against the murine *Asns* sequence results in a great amelioration of disease progression in orthologous PKD models (Fig. 8).

In a previous study, we had demonstrated that glutamine utilization in PKD is compensatory for the reduced import of pyruvate into mitochondria owned to the enhanced lactate production resulting from a Warburg-like effect (Podrini et al, 2018). In addition, we had shown that this compensatory enhanced utilization of glutamine is driven by the enzyme asparagine synthetase (ASNS) a transamidase which removes an aminic group from glutamine to add it to aspartate, thus releasing asparagine and glutamate as products (Lomelino et al, 2017). In our previous studies, we had demonstrated that silencing ASNS in vitro completely prevents the utilization of glutamine in the TCA cycle, demonstrating that glutamate production for the TCA cycle depends on ASNS in cells mutant for the *Pkd1* gene (Podrini et al, 2018) (Fig. 8). In addition, we had previously shown that in our cellular and animal models of PKD the classical glutaminase

enzyme GLS was not upregulated (Podrini et al, 2018), possibly explaining the inconsistent results obtained by different investigators when using GLS inhibitors (Flowers et al, 2018; Soomro et al, 2018). Here, we provide formal evidence that ASNS is upregulated in cystic tissues in vivo as well, while expression of this enzyme in wild-type tissues is extremely low. Of great interest, and in line with the model that we had proposed, a recent study found an increase in circulating asparagine in children and young adult PKD patients (Baliga et al, 2021), along with changes in the critical metabolites glutamine, glutamate and aspartate, products and substrates of the enzyme ASNS (Lomelino et al, 2017), providing strong supporting evidence of deregulation of this pathway in ADPKD patients. In the current study, we also show that changes in glutamine, glutamate, aspartate, and asparagine can be observed in the cystic PKD kidneys, and corrected upon silencing of ASNS in vivo, demonstrating that the changes in levels of circulating metabolites in the ADPKD population are likely the result of their change in the kidneys, possibly also in patients. Indeed, we show that ASNS is upregulated in the kidneys of all the ADPKD mouse models and human ADPKD kidney dataset analyzed, as well as in all our in vitro and in vivo models of PKD, both at the level of the transcript and of the protein.

This extensive evidence of ASNS upregulation in murine and in patients' tissues, along with our previous observation of the strong effect of *Asns* silencing in retarding growth in *Pkd1^−/−* MEFs provided the rationale for targeting ASNS in an orthologous mouse model of ADPKD, which mimics the slow progression of the pathology observed in patients. We have designed and developed a set of anti-*Asns*-ASOs that strongly decrease the expression of this enzyme in cystic kidneys (collaboration with Ionis Pharmaceuticals®, see "Methods"). Following in time the evolution of the main parameters characterizing the progression of PKD (i.e., increase in kidney volume and reduced renal function) revealed a quite striking decrease in kidney volume and weight, and a complete rescue of renal functionality down to the control levels upon administration of *Asns*-ASO as compared to controls (*Scr*-ASO). Of note, histological analysis of the *Asns*-ASO-treated kidneys revealed a quite variable phenotype not only between different animals, but also within each kidney with some samples presenting with portions of the kidneys that appeared completely rescued and others that appeared to remain somewhat cystic. This intra-sample variability was never observed in the controls and led us to propose that this effect is likely the result of a non-homogeneous distribution of the ASOs within the kidney as assessed by IHC. However, even when considering the great degree of variability of ASO distribution in the tissues, the improvement of disease manifestation and particularly of renal function was quite

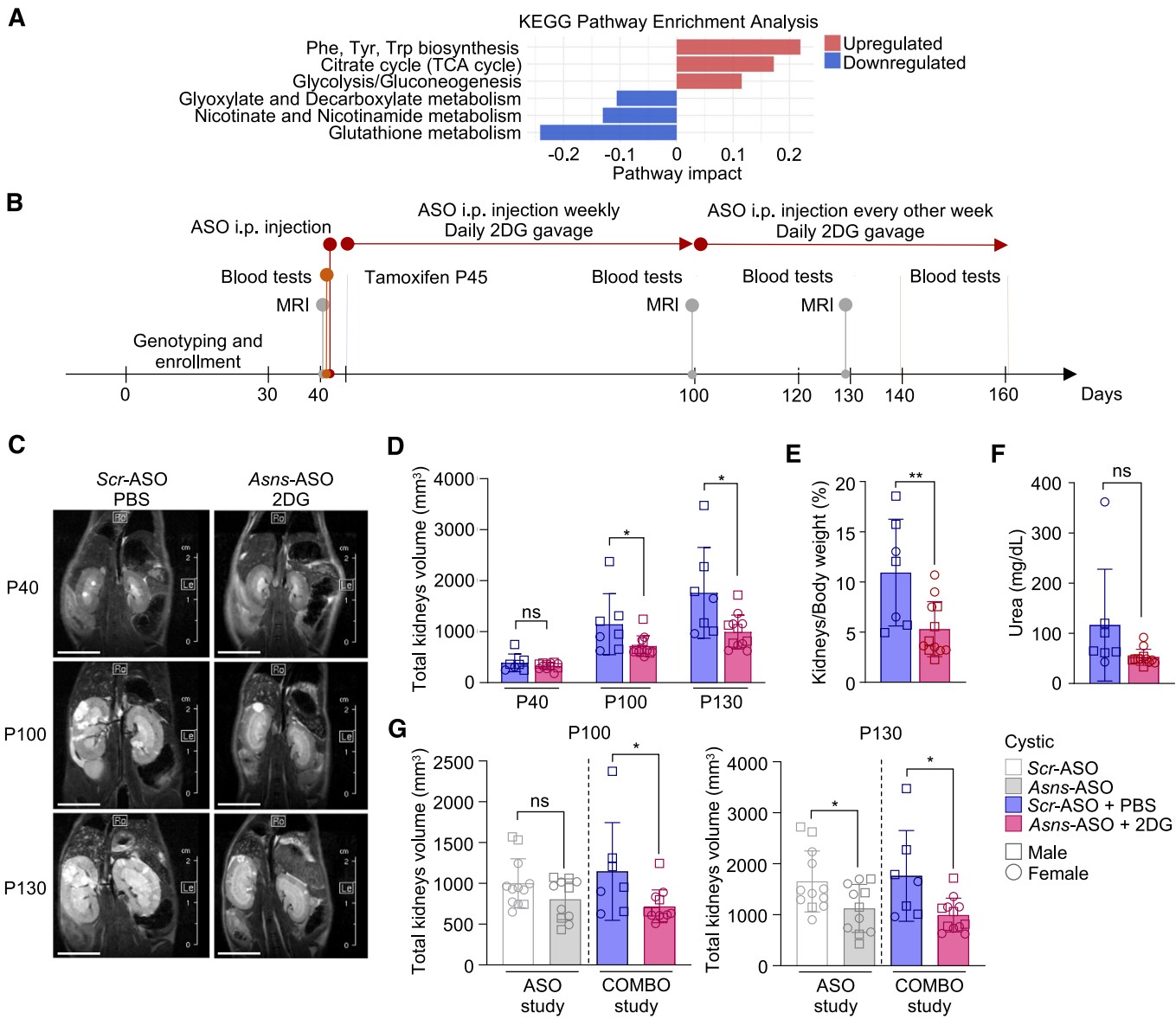

**Figure 7. Combined targeting of glutamine usage via *Asns* and glycolysis delays the PKD progression in *Pkd1^ΔC/f;*Tam-Cre* mice and ameliorates the endpoint phenotype and function.**

(A) Up- and downregulated Pathway Analysis Enrichment (cystic *Asns*-ASO not completely rescued compared to cystic *Scr*-ASO). (B) Experimental design of *Tam-Cre;Pkd1^ΔC/flox* and relative controls treated with *Asns*-ASO or *Scr*-ASO. (C) MRI representative images of cystic kidneys treated with *Scr*-ASO + PBS or *Asns*-ASO + 2DG. Images were acquired at P40 (before tamoxifen induction), P100 and P130. Scale bar (1 cm). (D) Total kidneys volume of cystic and control mice treated with *Scr*-ASO + PBS (*n* = 7, 5M-2F) or *Asns*-ASO + 2DG (*n* = 12, 5M-7F), calculated at P40, P100 and P130. (E) Percentage of kidneys weight normalized to total body weight of P160 mice treated with *Scr*-ASO + PBS (*n* = 7, 5M-2F) or *Asns*-ASO + 2DG (*n* = 12, 5M-2F). (F) BUN of cystic and control mice treated with *Scr*-ASO + PBS (*n* = 7, 5M-2F) or *Asns*-ASO + 2DG (*n* = 12, 5M-7F), measured P160. (G) Comparison of total kidneys volume measurement at P100 and P130 between cystic animals of the two presented studies: the single treatment with *Asns*-ASO (ASO study) and the combinatory treatment *Asns*-ASO + 2DG (COMBO study) (*n* = 12 cystic *Scr*-ASO; *n* = 11 cystic *Asns*-ASO; *n* = 7 cystic *Scr*-ASO + PBS; *n* = 12 cystic *Asns*-ASO + 2DG). ASO study bars correspond to data shown in Fig. 2E, while COMBO study bars correspond to data shown in (D). Data information: in A statistical significance of Pathway Enrichment and Impact was evaluated with Hypergeometric Test with relative-betweenness Centrality, based on KEGG Database with *P* < 0.05, FDR < 0.1. In (D–G) data are shown as mean ± SD. Unpaired two-tailed Student's *t* test. ns not significant. *P* < 0.05, **P* < 0.01. Source data are available online for this figure.

robust at the endpoint of the study. Of note, the study conducted in parallel on a group of controls treated with *Asns*-ASO revealed no indication of toxicity upon ASOs delivery, as assessed by body and kidney weight at sacrifice or overall appearance of the mice. These observations are relevant, because they show the potential to

translate the approach of ASNS inhibition to humans. Indeed, the expression levels of this enzyme are quite low in adult tissues, with the only exception of the exocrine pancreas, explaining why the silencing of *Asns* seems to be well tolerated. Further studies in this sense should be conducted in the future. It should be noted here

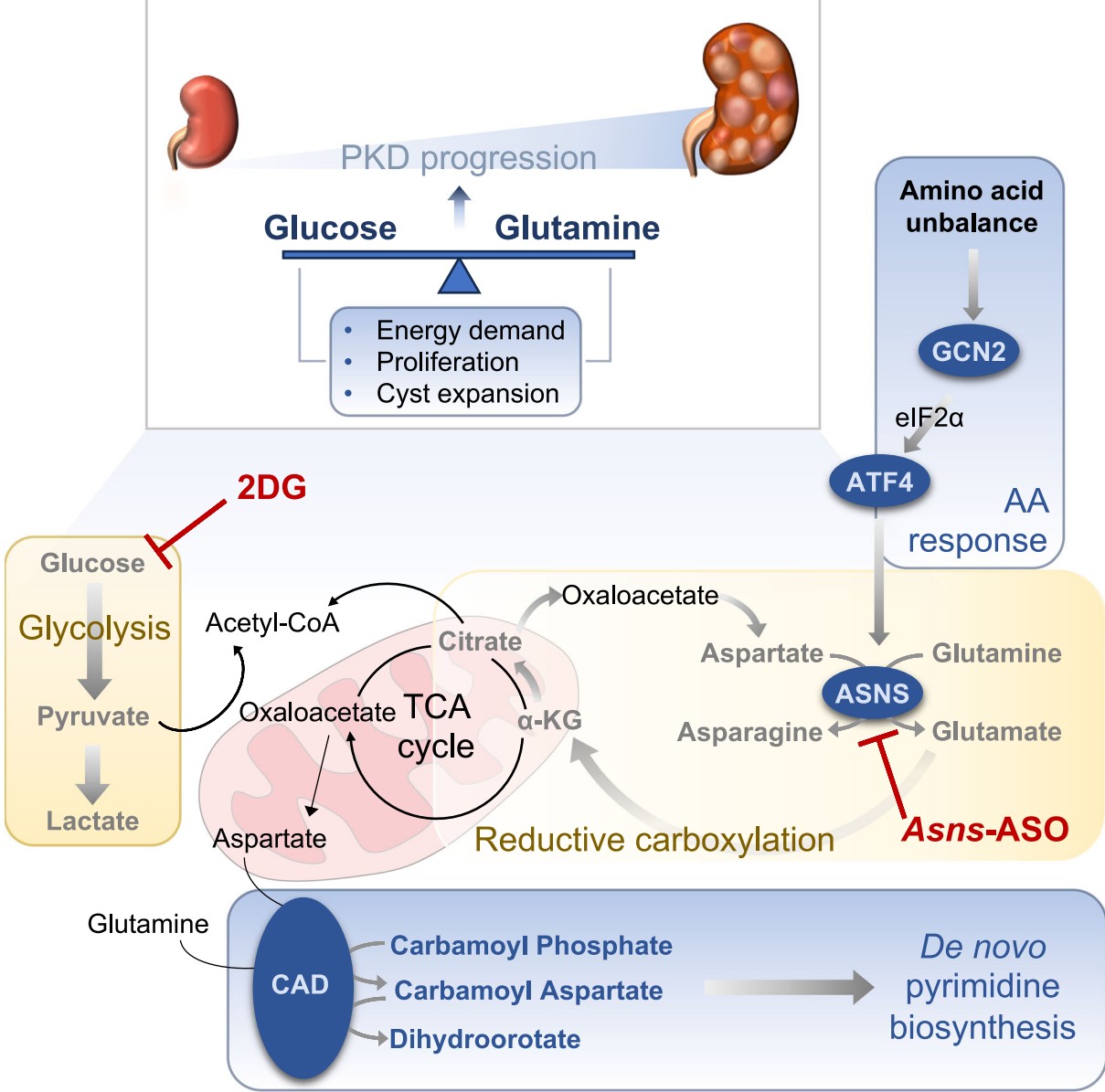

**Figure 8. Schematic representation of metabolic rewiring supported by ASNS in PKD and proposed strategy for therapy.**

The scheme summarizes how glucose and glutamine usage supports the progression of PKD. Increased glycolysis and lactate production (Rowe et al, 2013) reduce the fueling of glucose-derived carbons to the TCA cycle (Podrini et al, 2018) (yellow boxes). Glutamine utilization partially compensates for this deficit. Glutamine utilization is ASNS-dependent and fuels TCA cycle both through reductive carboxylation and oxidative phosphorylation (Podrini et al, 2018) (yellow boxes). Here we show that ASNS upregulation is downstream of the amino acid unbalance and GCN2-dependent AAR (blue box). In line with this, targeting *Asns* in vivo in PKD models retards disease progression. Metabolomic profiling in these tissues reveals a prominent upregulation of de novo pyrimidine biosynthesis as a consequence of ASNS enzyme upregulation and glutamine utilization (blue box). Furthermore, combining targeting of ASNS and glycolysis via treatment with ASO and 2DG, respectively, breaks the energetic balance acquired in PKD resulting in the amelioration of cystic phenotype. Bold arrows and text highlight processes found upregulated in PKD.

that targeting of ASNS has been proposed as a possible therapeutic approach, particularly in some solid tumors and in patients affected by acute lymphoblastic leukemia (Chiu et al, 2020). In this case, patients that tend to have high levels of circulating asparagine are subjected to treatment using asparaginase (ASNase) an enzyme able to degrade asparagine, which can be quite effective in some of these patients. However, some individuals develop resistance to the treatment due to an upregulation of the ASNS enzyme that can

bypass the ASNase treatment by increasing the production of asparagine, thus leading to resistance to therapy (Gutierrez et al, 2006). It has been proposed that in principle such patients would also benefit from inhibition of ASNS. The mechanism behind the involvement of ASNS in PKD, however, might be slightly different, as we have previously demonstrated that ASNS is required intracellularly for the generation of glutamate which fuels the TCA cycle (Podrini et al, 2018). The role of asparagine seems to be

less central in this disease, and in line with this observation, we have found that treatment with ASNase does not improve (and possibly exacerbates) disease progression. Thus, our data suggest that the production of glutamate (rather than that of asparagine) downstream of ASNS upregulation in PKD is central to disease progression and that the rationale for inhibiting ASNS might be quite different, though equally effective. One possible limitation of our studies stands in the fact that mice were treated with *Asns*-ASOs right from the beginning of disease initiation in the mouse, and it would be important to determine whether initiating treatment at later stages of disease manifestation is equally effective.

With respect to the mechanism leading to upregulation of ASNS, various previous studies have shown that while the protein is expressed at low levels in most tissues, it is strongly induced as a cellular response to cope with stress conditions, as part of activation of the integrated stress response (ISR). The role of ISR, mainly the UPR branch, in the context of cystic kidney disorders is still debated. Indeed, it has been described that activation of the PERK-dependent branch and activation of the downstream ATF4 transcriptional program contributes to establishment of a juvenile cystic kidney ciliopathy (Panda et al, 2022). On the other hand, Fedeles et al, demonstrated that activation of the UPR pathway plays a protective role against cyst formation, but that the pathway itself is not deregulated in those kidneys (Fedeles et al, 2015). Here, we demonstrated that the GCN2-dependent AAR branch of the ISR is strongly activated in ADPKD models, and that it drives the transcription of ASNS, since silencing of GCN2 in $Pkd1^{-/-}$ cells is sufficient to restore lower levels of ATF4 and ASNS (Fig. 8). Thus, our data imply that an alternative route of activation of the pathway, dependent on amino acid deregulation, can drive this response in PKD, in line with previous studies showing amino acids alteration in PKD (Ramalingam et al, 2021). But why is ASNS so essential for cyst growth and what are the consequences of its upregulation? Metabolomic profiling of ASO-treated kidneys sheds light on how ASNS can orchestrate multiple metabolic processes to support the progression of the pathology, and also provide evidence of how treatment could impact on this specific aspect. Despite the fact that we found several metabolic pathways deregulated in the PKD kidneys at P160 and that most of these are rescued by ASOs treatment, it should be noted that the strength of rescue can be very different from pathway to pathway, with some being completely restored back to controls and other being less positively affected by *Asns*-ASO. There could be multiple explanations for these findings. One possibility is that the pathways that are fully rescued are more closely related to the enzyme ASNS and to glutamine utilization and therefore are more robustly impacted. The second possibility is that some pathways have lower flexibility than others and cannot be compensated by carbon sources other than glutamine in PKD. In all cases, the non-homogeneous rescue of metabolic pathways seems to suggest that the metabolic rescue is not merely secondary to the rescue of disease progression, as in this case one would expect more homogeneous changes. Indeed, we think that this observation demonstrates that the *Asns*-ASO treatment effect on metabolic processes cannot be only reconducted to phenotypic rescue and restoration of healthy tissue. Importantly, the metabolomic profile corroborated the mechanism that supports the upregulation of ASNS in PKD. Indeed, we defined a profound unbalance in amino acid levels in PKD kidneys, which supports the transcriptional regulation of the enzyme via GCN2-dependent AAR activation.

The metabolomic profiling also provided important information on how ASNS targeting retards PKD progression. In a previous study we showed that PKD cells in vitro are more sensitive to reduced proliferation upon *Asns* silencing as compared to controls (Podrini et al, 2018). Here, we demonstrated that ASO treatment counteracts the upregulation of de novo pyrimidine biosynthesis in diseased kidneys, likely contributing to the reduced proliferation and consequent reduction in cyst expansion. Moreover, we defined an important role for ASNS in the proficient usage of glutamine in PKD cells, which can be used to sustain the CAD-dependent pyrimidine production, important for proliferation (Fig. 8). The purine biosynthesis pathway is also profoundly altered and corrected by *Asns*-ASO in vivo. The increased pyrimidine biosynthesis pathway and specifically the upregulation of orotate might further suggest that the enzyme dihydroorotate dehydrogenase (DHODH) might be tested as a target for therapy in future studies, even if one would in principle prefer to inhibit upstream effectors such as ASNS, which in addition represents a specific vulnerability of the diseased cells given the low levels of expression in wild-type cells, resulting in overall low toxicity.

Among the metabolic pathways that remain only partially corrected by *Asns*-ASO is that of glycolysis in line with the idea that the ASNS-mediated utilization of glutamine provides the carbon source for TCA cycle feeding, in conditions in which enhanced glycolysis depletes pyruvate import to mitochondria, but it is not the main provider of energy in these cells, which is almost entirely driven by glucose (Rowe et al, 2013; Podrini et al, 2018). Thus, given the effectiveness of the inhibitor of glycolysis 2-DG in improving PKD progression reported by multiple groups including ours (Chiaravalli et al, 2016), we tested whether combining the two strategies of interference with glucose utilization (by 2DG) and of glutamine utilization (by *Asns*-ASO) would act synergistically and provide the additional benefit that we would predict based on our in vitro data (Podrini et al, 2018). Indeed, we show that 2DG provides an additional improvement of disease amelioration in vivo over treatment with *Asns*-ASO alone. The effect is robust, but does not completely rescue disease progression, suggesting that additional ways of escape are available to the diseased PKD mutant cells.

One important question that remains unanswered is how the multiple metabolic alterations observed in ADPKD originate and what is the connection with the polycystins function. Recent studies have described that the cleaved C-tail of PC1 translocates to mitochondria and interacts with multiple metabolic enzymes (Lin et al, 2018, 2023; Onuchic et al, 2023; Pellegrini et al, 2023), although the molecular details of how these multiple interactions might result in the observed metabolic phenotype remain currently unexplained.

In this context, it should not be ignored that polycystins are ciliary proteins. In a recent study, we demonstrated that primary cilia respond to glutamine levels and that they do so via the enzyme ASNS, which localizes at the base of cilia (Steidl et al, 2023). In that case, we have shown that the removal of primary cilia from cells results in a metabolic reprogramming with respect to glucose and glutamine utilization, that is the opposite as compared to the one that we have observed in ADPKD, i.e., cells utilize reduced levels of glucose and reduced levels of glutamine. These data suggest that metabolic reprogramming might be part of the Cilia-Dependent Cyst Activating (CDCA) pathway, described to be a cascade

constitutively activated at cilia upon removal of the polycystins (Ma et al, 2013). While the molecular nature of this cascade remains obscure, the existence of such a pathway seems to be demonstrated by the fact that the removal of cilia unexpectedly improves disease progression in the context of *Pkd1* or *Pkd2* mutants (Ma et al, 2013). Here, we propose that ASNS might be a key component of this CDCA pathway. Indeed, this enzyme seems to respond to all criteria for a molecular definition of the CDCA pathway: (i) it is upregulated in the absence of the polycystins (Podrini et al, 2018), but downregulated in the absence of cilia (Steidl et al, 2023); (ii) it localizes at primary cilia (Steidl et al, 2023); (iii) its inhibition results in renal cysts improvement (Current work). Further studies should be undertaken to better understand this aspect, while the main conclusion of our current work is that we have identified a novel and quite robust new target for therapy in ADPKD, the enzyme ASNS.

## Study limitations

The present study has also multiple limitations that will need to be addressed in the future: (i) the treatment was performed in mice as kidneys were developing cysts, it will be important to test the effect on relatively advanced disease conditions; (ii) we do not fully understand the mechanism responsible for retarding disease progression and in particular the role of reduced proliferation; (iii) we do not know whether the metabolic changes observed are primary events upon inactivation of the *Pkd1* gene or only manifest after cysts have formed.

# Methods

## Cell culture and treatment

Immortalized $Pkd1^{+/+}$ and $Pkd1^{-/-}$ MEFs (Distefano et al, 2009) were cultured in high-glucose Dulbecco's modified Eagle medium DMEM (Thermo Fisher Scientific, #41965062), supplemented with 10% fetal bovine serum (FBS), 1% penicillin-streptomycin (Pen-Strep; Thermo Fisher Scientific, #15070-063). Cells were thawed at passage 36 and kept in culture for experiments for a maximum of ten passages.

The mouse cortical collecting duct (mCCD) parental cell line was Kindly provided by Dr. Eric Féraille (University of Lausanne, Switzerland) (Montesano et al, 2009; Gaeggeler et al, 2005), and cultured in the following growth medium: DMEM-F12 supplemented with 5 μg/mL of insulin, 60 nM of selenium, 5 μg/mL of transferrin, 5 ng/mL of mouse EGF, 1 nM of triiodothyroidine, 50 nM of dexamethasone, and 2% of decomplemented FBS.

To generate monoclonal isogenic *Pkd1* KO mCCD cells, we employed U6gRNA-Cas9-2A-GFP plasmids (Sigma-Aldrich) carrying three distinct custom-designed guide RNA (gRNA) sequences targeting exon 1 (gRNA#1: CCCTGCTTTTGCGGCCCTGCGC; gRNA#2: CCGCCTGCCGCGTCAATTGCTC; gRNA#3: CCGCGTCAATTGCTCCGGCCGC). The CMV-Cas9-2A-RFP-scrambled gRNA plasmid was used as a control. Cells were plated on 100-mm$^2$ plates the day before the transfection. Transfection was performed using Lipofectamine 2000 (Thermo Fisher Scientific, #11668019) following the manufacturer's instructions, 5 μg of plasmid DNA per dish with 1:3 DNA/Lipofectamine ratio were used.

Three days after transfection cells were sorted by FACS for GFP (potential *Pkd1* KO mCCD cells) or RFP (control mCCD cells) and plated as single cells into 96-well plates. The vital clones were sequentially expanded and screened for PC1 protein expression by western blot. We obtained 6 control clones and 6 *Pkd1* KO clones, and we selected 2 Ctrl clones and 5 *Pkd1* KO clones obtained by guides #2 or #3 for further analysis. Cells were cultured in DMEM supplemented with 10% FBS (diet medium) for at least 3 weeks before experiments were performed starting from passage 25 to passage 32.

For glucose deprivation, MEF cells were plated in DMEM 10% FBS in 100-mm plates (200,000 $Pkd1^{+/+}$ cells and 400,000 $Pkd1^{-/-}$ cells). After 8 h, cells were subjected to glucose deprivation for 24 h.

For glucose and serum deprivation, pools of control and $Pkd1^{-/-}$ mCCD cells were seeded in diet medium ($2 \times 10^6$ or $4 \times 10^6$ cells/100-mm plate, low- and high confluency, respectively). After 36 h, cells were subjected to glucose and serum deprivation for 24 h.

For the rapamycin time course, MEF cells were plated in 60-mm plates (400,000 $Pkd1^{+/+}$ cells and 200,000 $Pkd1^{-/-}$ cells). The day after, cells were subjected to 6 h of serum starvation before 50 nM rapamycin treatment for 4, 8, or 24 h. No authentication was performed. All cell lines were tested negative for mycoplasma contamination.

## siRNA

For transient silencing of *Asns* (Ambion, #AM16704/188316) and *Eif2ak4*, 20 nM predesigned siRNA (Ambion, #4390815/s77548) have been used along with scrambled control (Ambion, #AM4613), following the manufacturer's instructions. MEF cells were plated in six-well plates (130,000 $Pkd1^{+/+}$ cells and 180,000 $Pkd1^{-/-}$ cells) for siRNA transfection, in DMEM supplemented with 10% dialyzed FBS (Thermo Fisher, #26400-044), without antibiotics. Transfection has been performed for 2 consecutive days, reaching a final concentration of 30 nM siRNA, using Lipofectamine RNAiMAX (Thermo Fisher, #13778150), following the manufacturer's instructions. Cells have been processed for RNA and protein extraction after 48 h from the first transfection.

## Ultra-high-pressure liquid chromatography–mass spectrometry (MS) tracing metabolomics experiments

Tracing metabolomics experiments on $Pkd1^{-/-}$ and control MEF cells has been conducted as previously described (Podrini et al, 2018). Briefly, cells were cultured in either culture media enriched with $^{13}C_5$-glutamine (Cambridge Isotope Laboratories, CLM-1822-H-PK) or $^{15}N_2$-glutamine (Cambridge Isotope Laboratories, NLM-1328-PK). Metabolite assignments, $^{13}C_5$-glutamine and $^{15}N_2$-glutamine tracing experiments, isotopologue distributions, and correction for expected natural abundances of 13C and 15N isotopes, were performed using MAVEN (Princeton, NJ, USA) (Clasquin et al, 2012) and RStudio AccuCor (Su et al, 2017). Metabolite levels were then normalized to the delta of BCA protein quantification (Pierce™ BCA Protein Assay Kits 23225) between cells at 24 h after tracing and at t0.

## Generation of Tam-Cre;Pkd1$^{ΔC/flox}$ and KspCre;Pkd1$^{ΔC/flox}$ mice

C57/BL6N *Tam-Cre;Pkd1$^{ΔC/+}$* were crossed with C57/BL6N *Pkd1$^{flox/flox}$* to generate the *Tam-Cre;Pkd1$^{ΔC/flox}$* mice (Chiaravalli et al, 2016). Cre

recombinase activity was induced by a single injection of Tamoxifen (250 mg/kg; Sigma-Aldrich, #T5648) at P45 or P25 for a long-term model involved in *Asns*-ASO study and the previous pilot study, respectively. Tamoxifen was freshly prepared and dissolved in corn oil (#C8267; Sigma-Aldrich) by continuous shaking at 37 °C for at least 6 h and was injected intraperitoneally. For the generation of *KspCre;Pkd1^{ΔC/flox}*, we crossed C57/BL6N *Pkd1^{flox/flox}* with *KspCre;Pkd1^{ΔC/+}*, as previously described (Rowe et al, 2013). Animal care and experimental protocols were approved by the Institutional Care and Use Ethical Committee at the San Raffaele Scientific Institute and further approved by the Italian Ministry of Health (IACUC 736 and IACUC 921).

## Treatment with ASOs in vivo

*Tam-Cre;Pkd1^{ΔC/flox}* mice and relative controls were randomly distributed in the group of treatment for each study. Given that sporadic cysts due to ambient tamoxifen could be observed in the liver or kidney, we measured kidney volume prior to tamoxifen injection and mice with kidney volumes significantly higher than matching controls during the first MRI scan represented exclusion criteria for the study. After enrollment, mice were treated with antisense oligonucleotide targeting *Asns* (*Asns*-ASO; TATTTTATCACACTCC) or non-targeting scramble control (*Scr*-ASO; ACGATAACGGT-CAGTA) (Ionis Pharmaceuticals®). As the efficacy/tolerability was dosed at 50 mg/kg/week, ASOs were administered at the same dosage via weekly intraperitoneal injection for the first 2 months after Tamoxifen induction, and every second week after P100 untill the end of the experiment. In the pilot experiment, ASOs were administered weekly at the same dosage till P60 and every second week until the end of the study. For the combinatory approach, experimental animals were treated daily (5 days per week) with 100 mg/kg 2DG (Sigma-Aldrich, #D8375) gavage (combined with *Asns*-ASO treatment) or PBS gavage (combined with *Scr*-ASO treatment).

During all the studies presented, animals were subjected to five blood tests (P40, P100, P130, P140, P160) to monitor renal and liver parameters and 3 MRI scans (P40, P100, P130) to monitor kidney volume.

At the end of the study, animals were weighed and perfused with cold PBS after anesthesia. Kidneys were collected and weighed to calculate kidneys/body weight. Tissues were then processed for ex vivo analysis. The treatment studies were not conducted in blind, MRI acquisition and volume calculations were conducted blindly, and the biochemical analysis was performed blindly by a facility (see below).

## Biochemical serum analysis

Urea (#0018255440) and Creatinine (#0018255540) were used for the quantitative determination of the serum level with an International Federation of Clinical Chemistry and Laboratory Medicine optimized kinetic ultraviolet (UV) method in an ILab650 chemical analyzer (Instrumentation Laboratory). Urea and Crea are expressed as mg/dl. SeraChem Control Level 1 and Level 2 (#0018162412 and #0018162512) were analyzed as quality control.

## MRI

All MRI studies were performed on a 7-T Preclinical Scanner (BioSpec 70/30 USR, Paravision 6.0.1; Bruker) equipped with 450/ 675 mT/m gradients (slew rate: 3400–4500 T/m per second; rise time: 140 μs) and a circular polarized mouse body volume coil with an inner diameter of 40 mm. Mice were under general anesthesia obtained by 1.5–2% isoflurane vaporized in 100% oxygen (flow of 1 L/min). Breathing and body temperature were monitored during MRI (SA Instruments, Inc., Stony Brook, NY) and maintained around 30 breaths per minute and 37 °C, respectively. All MRI studies included axial and coronal RARE T2-weighted sequences (slice thickness, 0.6 mm; interslice gap, 0 mm) with several slices, allowing complete coverage of both kidneys. Manual segmentation of the kidneys was performed on each slice using NIH software MIPAV (version 7.4.0), excluding the collecting system, and kidney volume results from automatic summation of voxel volumes. Both acquisition and analysis of the data were performed blindly by an operator unaware of genotype or treatment conditions.

## Histology and immunohistochemistry

After euthanasia, kidneys were collected, washed in PBS, weighed, and fixed in 10% neutral buffered formalin containing 4% formaldehyde (BioOptica, 05-01V15P). Tissues were transferred to 70% ethanol after 24 h, and paraffin-embedded for subsequent analysis. Formalin-fixed paraffin-embedded consecutive sections (4 μm) were dewaxed and hydrated through graded decrease alcohol series and stained for histology or immunohistochemical characterization (IHC).

For ASO detection, slides were stained with Rabbit polyclonal ASO (Ionis) antibody on a Ventana Ultra staining system. ASO slides were treated enzymatically with Trypsin (Sigma, T8003). The slides were then blocked with an Endogenous Biotin Blocking Kit (Ventana, 760-050) and Normal Goat Serum (Jackson Immuno Labs, 005-000-121). The primary antibody was diluted with Discovery Antibody Diluent (Ventana, 760-108) and incubated for 1 h at 37 °C. The antibodies were detected with Biotin labeled Goat Anti-Rabbit secondary antibody (Jackson Immuno Labs, 111-005-003). The secondary antibody was labeled with DABMap Kit (Ventana, 760-124). Images were scanned on a Hamamatsu S360 scanner at ×20 resolution.

For histological analysis in bright-field microscopy, slides were stained using standard protocols for Hematoxylin and Eosin (using Mayer's Hematoxylin, BioOptica #05-06002/L and Eosin, BioOptica #05-10002/L).

For IHC characterizations, slides were immunostained with Automatic Leica BOND RX system (Leica Microsystems GmbH, Wetzlar, Germany). First, tissues were deparaffinized and pre-treated with the Epitope Retrieval Solution (ER1 Citrate Buffer) at 100 °C. Primary antibody against Ki-67 ((D3B5) Rabbit mAb (Mouse Preferred; IHC Formulated) (CST, #12202)) was used 1:200 and developed with Bond Polymer Refine Detection (Leica, DS9800). Primary antibody against ASNS (CST, #92479) was used 1:250 and developed with Bond Polymer Refine Detection (Leica, DS9800).

Slides were acquired with Aperio AT2 digital scanner at a magnification of ×20 (Leica Biosystems) and analyzed with Imagescope (Leica Biosystem).

## Cystic index

Images of transversal sections from the inner part of the kidney after hematoxylin–eosin staining were taken (Zeiss AxioImager M2m with

AxioCam MRc5) at ×2.5 magnification. The ImageJ program (http://rsb.info.nih.gov/ij/) was then used to quantify the total surface of the kidney and the total cystic area. We next calculated the ratio between the cystic area and the total area of the kidney. Histological analysis and cystic index calculations were not conducted blindly.

## Ki-67 quantification

Proliferative Ki-67-positive cells have been quantified semi-automatically on ×10 magnification images. Ki-67 nuclear staining has been automatically detected in the epithelium lining cortical cysts by Aperio Image analysis software (Leica Biosystems). Positive nuclei have been manually counted and normalized on the total number of nuclei in cyst epithelium.

## Ultra-high-pressure liquid chromatography–mass spectrometry (MS) metabolomics experiments

Kidney tissues were ground in dry ice and weighed. In all, 15 mg of powder was extracted in 1 mL of ice-cold extraction solution (methanol:acetonitrile:water 5:3:2 v/v/v) (Nemkov et al, 2017). Suspensions were vortexed continuously for 30 min at 4 °C. Insoluble material was removed by centrifugation at $18,000 \times g$ for 10 min at 4 °C and supernatants were isolated for metabolomics analysis by UHPLC-MS.

Analyses were performed using a Vanquish UHPLC coupled online to a Q Exactive mass spectrometer (Thermo Fisher, Bremen, Germany). Ten microliters of sample extracts were loaded onto a Kinetex XB-C18 column ($150 \times 2.1$ mm i.d., 1.7 μm—Phenomenex). Samples were analyzed using a 5 min method as described (Nemkov et al, 2017; Reisz et al, 2019).

Metabolite assignments were performed using MAVEN (Princeton, NJ, USA) (Clasquin et al, 2012). Metabolites levels were then normalized to BCA protein quantification (Pierce™ BCA Protein Assay Kits 23225). Graphs and statistical analyses (either PCA, HCA or MetPA) were prepared with GraphPad Prism 10.0 (GraphPad Software, Inc, La Jolla, CA), GENE-E (Broad Institute, Cambridge, MA, USA), MetaboAnalyst 5.0 (Chong et al, 2018), and RStudio Team (2020). RStudio: Integrated Development for R. RStudio, PBC, Boston, MA—URL http://www.rstudio.com/.

## Real-time PCR analysis

RNA was isolated from plated cells or snap-frozen kidneys using the RNAspin Mini kit (GE Healthcare, #25-0500-72) according to the manufacturer's instructions. For reverse transcription of RNA, either Oligo(dT)15 primers (Promega, #c1101) or Random primers (Promega, #c1181) and ImProm-II Reverse Transcriptase (Promega, #A3802) were used. Quantitative real-time PCR analysis was performed on duplicates or triplicates using SYBR Green I master mix (Bio-Rad, 48873) on CFX96 Real-Time PCR Instrument (Bio-Rad). Primers sequences:

|         | Forward                      | Reverse                      |
|---------|------------------------------|------------------------------|
| *Asns*  | GGTTTTCTCGATGCCTCCTT         | TGTGGCTCTGTTACAATGGTG        |
| *Eif2ak4* | CCACGAGATTCAGAGAAGGAAAG    | TGGGTCTCTCTTAGAGGCATAG       |
| *Hprt*  | CACAGGACTAGAACACCTGC         | GCTGGTGAAAAGGACCTCT          |

## Western blot analysis

Kidneys, grinded on dry ice, or cells were lysed in lysis buffer solution (150 mM NaCl (Sigma-Aldrich, #s9625), 20 mM Na2HPO4 (BDH, #10494L)/NaH2PO4 (BDH, #102455S), 10% glycerol (Sigma-Aldrich, #G7757), 1% Triton X-100 (pH 7.2), complete protease inhibitor cocktail (Roche, #11836145001) and phosphatase inhibitors (1 mM final concentration of glycerophosphate (Sigma-Aldrich, #G9891), sodium orthovanadate (Sigma-Aldrich, #S6508) and sodium fluoride (Sigma-Aldrich, #S6521)). For immunodetection of PC1, before lysis, cells were detached from cell culture dish on ice with the cell scraper, collected and harvested by centrifugation. Cells were resuspended in PBS, subjected to four cycles of freeze & thaw, and then harvested by centrifugation. Total protein extracts were quantified and Laemmli buffer was added to the samples. Proteins were resolved in 3–8% Tris-Acetate gels (Life Technologies, Carlsbad, CA, USA) and transferred onto nitrocellulose membranes (Millipore, Merck KGaA, Darmstadt, Germany). Next, we used 5% skim milk in Tris-buffered saline and Tween-20 (TBS-T) (Sigma-Aldrich, P1379) for blocking. Primary antibodies were diluted in 1× TBS-T supplemented with 3% of BSA (Sigma-Aldrich, #A7906). HRP-conjugated secondary antibodies were diluted 1:5000 (anti-mouse, Life Technologies, #A11003; anti-Rabbit, Life Technologies, #11035) (or more if necessary) in 5% of skim milk, 1× TBS-T, and detected by ECL (Cytiva, #RPN2106) alone or supplied with 10% of SuperSignal West Femto (Thermo Fisher Scientific, #34095) when required, or Clarity Western ECL substrate (Biorad, #1705060), or Radiance Plus ECL (Azure Biosystems, #AC2103).

| Antibody | Dilution | Manufacturer |
|----------|----------|--------------|
| ASNS | 1:500 | Santa Cruz Biotechnology, #sc-365809 |
| ATF4 | 1:500 | Santa Cruz Biotechnology, #sc-390063; Cell Signalling Technologies, #11815 |
| CAD | 1:1000 | Cell Signalling Technologies, #11933 |
| GCN2 | 1:500 | Cell Signalling Technologies, #3302 |
| P-CAD | 1:1000 | Cell Signalling Technologies, #12662 |
| P-GCN2 | 1:500 | Cell Signalling Technologies, #94668 |
| Polycystin-1 C-terminal fragment (E8) | 1:500 | From PKD-RRC, https://www.pkd-rrc.org |
| Polycystin-1 N-terminal fragment (7E12) | 1:500 | Santa Cruz Biotechnology, TX, USA, #sc-130554 |
| Vinculin | 1:5000 | Sigma-Aldrich, #05-386 |

## Immunofluorescence

For immunofluorescence (IF) analysis, MEF cells were plated on glass coverslips. Silencing of *Asns* was performed as previously described. After 48 h from the first transfection, cells were fixed for 10 min in 4% paraformaldehyde (PFA) (Electron Microscopy Sciences, #157-4) followed by permeabilization in 0.1% Triton X-100 (Sigma-Aldrich, #T8787) in PBS. After 1 h blocking in 3% BSA (Sigma-Aldrich, #A7906) in PBS at Room Temperature (RT),

## The paper explained

### Problem

Autosomal dominant polycystic kidney disease (ADPKD) is one of the most common monogenic disorders affecting humans. The disease manifests with the formation of cysts in both kidneys. Cysts are enclosed structures of epithelia that grow over time leading to the compression of the surrounding normal parenchyma, eventually causing loss of renal function. Only one compound, Tolvaptan, able to slow down growth and retard progression is currently available to patients, but it presents with limited efficacy and limited tolerability. We and others described a peculiar role of metabolic reprogramming in PKD which favors growth of the cysts and have proposed that this cellular dysfunction represents an important vulnerability to be targeted for therapy.

### Results

Here we used antisense oligonucleotides (ASOs) to reduce the expression levels of the metabolic enzyme asparagine synthetase (ASNS) in orthologous and slowly progressive ADPKD murine models. We report that treatment with such ASOs greatly improves disease progression and restores almost normal function in mice. The effect of silencing prominently improves the metabolic dysfunction in PKD tissues and identifies novel metabolic vulnerabilities. We also show that combining a glycolysis inhibitor (2DG) with silencing of ASNS further ameliorates disease manifestation.

### Impact

We have identified a novel target for therapy in Polycystic Kidney Disease, whose targeting results in great amelioration of disease progression and regression of several metabolic alterations. The results might lead to the development of a much-needed novel therapy for this disorder.

incubation with anti-ASNS primary antibody (CST, #92479) diluted 1:125 in 3% BSA in PBS was performed out overnight at 4 °C. Incubation with secondary antibody diluted in 3% BSA in PBS was carried out for 1 h at RT and nuclei were stained with DAPI. Glasses were then mounted with Fluorescence Mounting medium (Dako, #S3023). Images were obtained using Zeiss Axio Observer.Z1 microscope.

## Datasets sources

Normalized counts were obtained directly from publications (Olson et al, 2019), normalized probe intensity data were extracted from the series matrix files from the GEO repository with accession numbers: GSE7869 (Song et al, 2009) and GSE121563 (Podrini et al, 2018). Microarray probes of each dataset were aggregated on a gene basis by averaging normalized probe intensities.

## Data visualization

scRNA-seq dotplots visualizing the expression of controls and ADPKD patients were created using the "Kidney Interactive Transcriptomics" platform http://humphreyslab.com/SingleCell (Wu et al, 2022). Expression heatmaps of RNA-seq and Microarray expression data were created using R (R Core Team 2022, https://www.R-project.org/) and the ComplexHeatmap package (Gu et al, 2016).

## Statistical analysis

Statistical analyses were performed using Prism 10, GraphPad Software. *t* test was used for all analyses comparing two groups. ANOVA statistical analysis followed by Tukey's multiple comparison test was performed in all analyses where more than two groups were present.

For in vivo studies, animals were randomly distributed in experimental groups, and ASO treatments were not conducted in blind. MRI acquisition and volumes calculations were conducted blindly, and the biochemical analysis was performed blindly by a facility.

## For more information

Boletta Lab website: https://research.hsr.it/en/divisions/genetics-and-cell-biology/cystic-kidney-disorders.html. Associazione Italiana Rene Policistico (AIRP) website: https://www.renepolicistico.it/. OMIM #601313: https://omim.org/entry/601313. PKD Foundation website: https://pkdcure.org. PKD International: https://www.pkdinternational.org.

## Data availability

The mass spectrometry metabolomics data are available at the NIH Common Fund's National Metabolomics Data Repository (NMDR) website, the Metabolomics Workbench, https://www.metabolomicsworkbench.org where it has been assigned StudyID ST003111 (datatrack_id:4677 and datatrack_id:4679). The data can be accessed directly via its Project https://doi.org/10.21228/M8VM8W. The private link for datatrack 4679 is: http://dev.metabolomicsworkbench.org:22222/data/DRCCMetadata.php?Mode=Study&StudyID=ST003113&Access=IfnA6608. The DOI for this project (PR001933) is: https://doi.org/10.21228/M8VM8W. The private link for datatrack 4677 is: http://dev.metabolomicsworkbench.org:22222/data/DRCCMetadata.php?Mode=Study&StudyID=ST003111&Access=TlqE6018. The DOI for this project (PR001933) is: https://doi.org/10.21228/M8VM8W.

The source data of this paper are collected in the following database record: biostudies:S-SCDT-10_1038-S44321-024-00071-9.

## Peer review information

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

## Acknowledgements

The authors are grateful to the other members of the Boletta laboratory for useful discussions. Technical help was provided by the core facility of Animal Histopathology, in particular A. Fiocchi, and the Animal Biochemistry core facility, in particular M. Raso and M. Ravà. This work was supported by the Italian Ministry of Health (RF-2018-12368254 to AB; GR-2016-02364851 to CP), the Italian Association of Patients with PKD (AIRP to AB), the Italian Association for Research on Cancer (AIRC, IG2019-23513 to AB). The authors are grateful to Dr. Silvia Bramani for her continuous support.

## Author contributions

**Sara Clerici**: Conceptualization; Data curation; Formal analysis; Investigation; Writing—original draft; Writing—review and editing. **Christine Podrini**: Conceptualization; Data curation; Formal analysis; Methodology; Writing—original draft. **Davide Stefanoni**: Conceptualization. **Gianfranco Distefano**: Data curation; Investigation. **Laura Cassina**: Data curation; Validation; Methodology. **Maria Elena Steidl**: Data curation; Formal analysis; Methodology. **Laura Tronci**: Data curation; Formal analysis; Methodology. **Tamara Canu**: Data curation; Methodology. **Marco Chiaravalli**: Data curation; Formal analysis; Investigation; Methodology. **Daniel Spies**: Data curation; Formal analysis. **Thomas A Bell 3rd**: Resources; Formal analysis; Methodology. **Ana SH Costa**: Data curation; Formal analysis; Methodology. **Antonio Esposito**: Supervision. **Angelo D'Alessandro**: Supervision. **Christian Frezza**: Supervision; Methodology. **Angela Bachi**: Supervision; Methodology. **Alessandra Boletta**: Conceptualization; Data curation; Supervision; Funding acquisition; Investigation; Writing—original draft; Project administration; Writing—review and editing.

Source data underlying figure panels in this paper may have individual authorship assigned. Where available, figure panel/source data authorship is listed in the following database record: biostudies:S-SCDT-10_1038-S44321-024-00071-9.

## Disclosure and competing interests statement

AB, CP, and MC are co-inventors on patents related to metabolic interventions in Polycystic Kidney Disease including one on 2DG use in PKD and one on the silencing of ASNS for PKD. The remaining authors declare no competing interests.

# Expanded View Figures

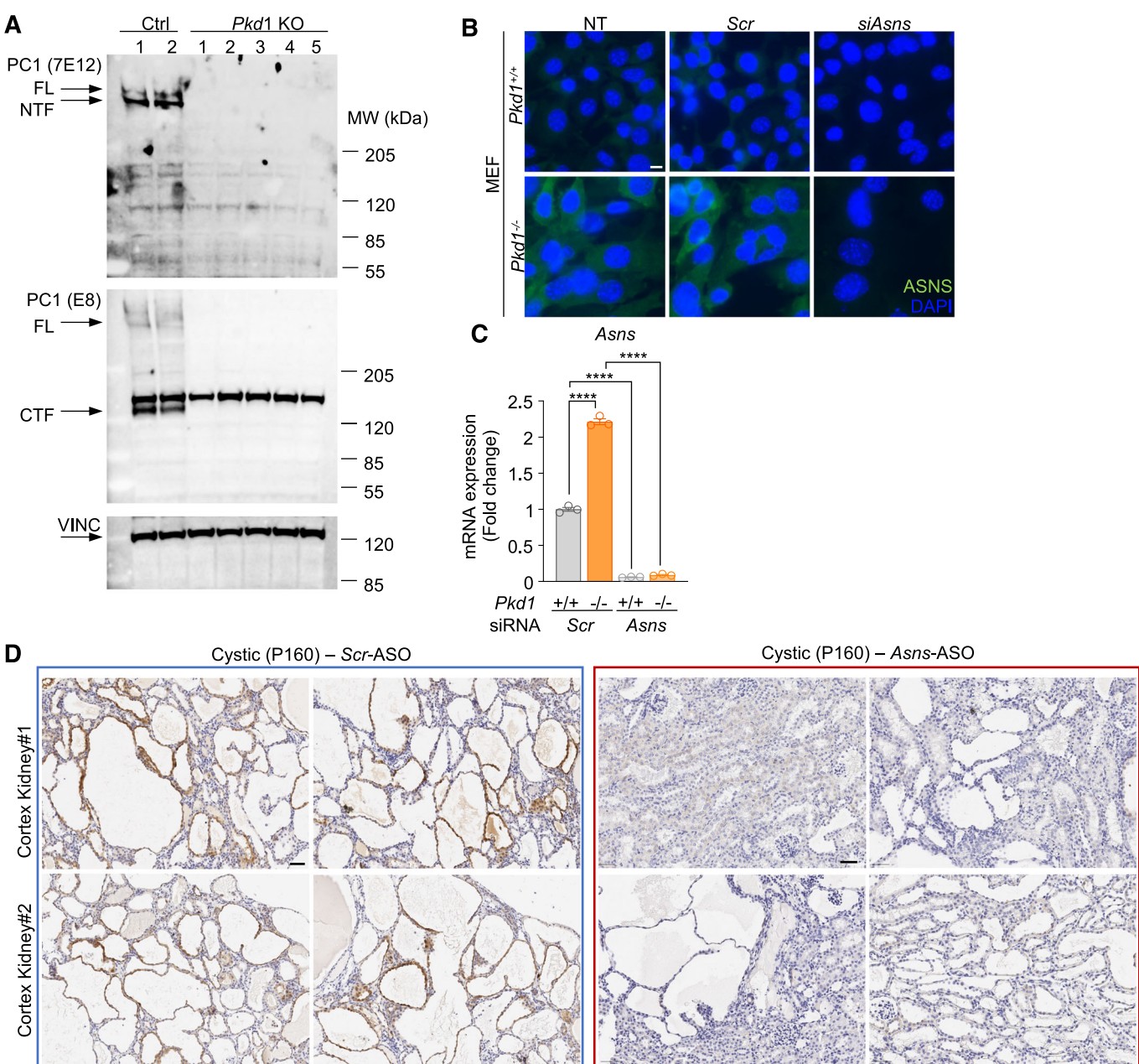

**Figure EV1. Validation of cell lines KO for Pkd1 and of antibodies directed against ASNS.**

(A) Immunoblotting on control and *Pkd1* KO mCCD clones for the detection of PC1 protein, using PC1 7E12 and E8 antibodies. FL, PC1 full-length protein; NTF, PC1 N-terminal fragment; CTF, PC1 C-terminal fragment. PC1 bands were detected in control cells and not in *Pkd1* KO clones. Vinculin was used as loading control to show equal loading of protein samples. (B) Representative images of immunofluorescence staining of ASNS in *Pkd1*$^{-/-}$ and *Pkd1*$^{+/+}$ MEF cells, untreated (NT), scrambled (*Scr*) or silenced for *Asns*. Scale bar (10 µm). (C) *Asns* expression in *Pkd1*$^{-/-}$ and control MEF cells upon silencing (*n* = 3 biological replicates). (D) Representative images of IHC ASNS staining in cystic renal epithelium of *Tam-Cre;Pkd1*$^{\Delta C/flox}$ (P160) mice treated with *Scr*-ASO or *Asns*-ASO. Scale bar (50 µm). Data information: In (C) data are shown as mean ± SD. One-way ANOVA, corrected with Tukey's multiple comparisons. ****$P < 0.0001$.

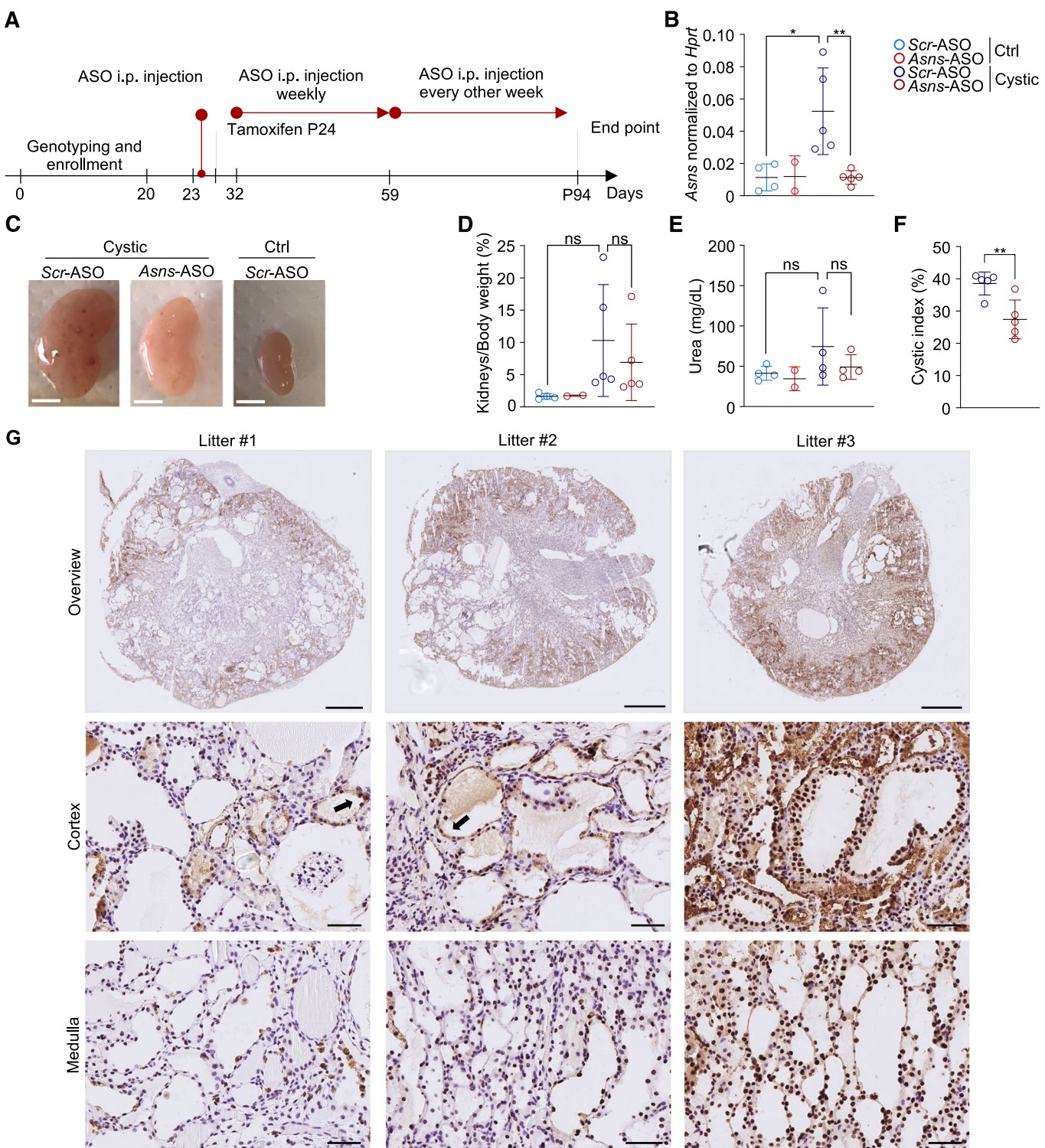

**Figure EV2.   Pilot test of Asns-ASO on a medium-term PKD model.**

(A) Experimental design of pilot study on *Tam-Cre;Pkd1*<sup>ΔC/flox</sup> and relative controls treated with *Asns*-ASO or *Scr*-ASO ($n = 4$ ctrl *Scr*-ASO; $n = 2$ ctrl *Asns*-ASO; $n = 5$ cystic *Scr*-ASO; $n = 5$ cystic *Asns*-ASO). (B) *Asns* mRNA expression in *Tam-Cre;Pkd1*<sup>ΔC/flox</sup> and relative controls treated with *Asns*-ASO or *Scr*-ASO ($n = 4$ ctrl *Scr*-ASO; $n = 2$ ctrl *Asns*-ASO; $n = 5$ cystic *Scr*-ASO; $n = 5$ cystic *Asns*-ASO). (C) Representative images of cystic kidneys and relative controls at P94 treated with *Asns*-ASO or *Scr*-ASO. (D) Percentage of kidneys weight normalized to body weight of cystic and relative controls treated with *Asns*-ASO or *Scr*-ASO ($n = 4$ ctrl *Scr*-ASO; $n = 2$ ctrl *Asns*-ASO; $n = 5$ cystic *Scr*-ASO; $n = 5$ cystic *Asns*-ASO). (E) BUN of cystic and relative control kidneys treated with *Asns*-ASO or *Scr*-ASO ($n = 4$ ctrl *Scr*-ASO; $n = 2$ ctrl *Asns*-ASO; $n = 4$ cystic *Scr*-ASO; $n = 4$ cystic *Asns*-ASO). (F) Quantification of the cystic area percentage of the total kidney area measured in transversial sections of cystic *Scr*-ASO or *Asns*-ASO groups ($n = 5$ cystic *Scr*-ASO; $n = 5$ cystic *Asns*-ASO). (G) ASO distribution in cystic kidneys harvested from *Asns*-ASO-treated mice at P95 from three different litters. Representative images of ASO distribution in total kidneys (upper panel, scale bar (1 mm)), renal cortex middle panel, scale bar (50 μm)), and renal medulla (lower panel, scale bar (50 μm)). Arrows indicate the ASO-positive cystic epithelium. Data information: in (B, D, E) data are shown as mean ± SD. One-way ANOVA. ns non-significant; *$P < 0.05$; **$P < 0.01$. In (F) data are shown as mean ± SD. Student's unpaired two-tailed *t* test. **$P < 0.01$.

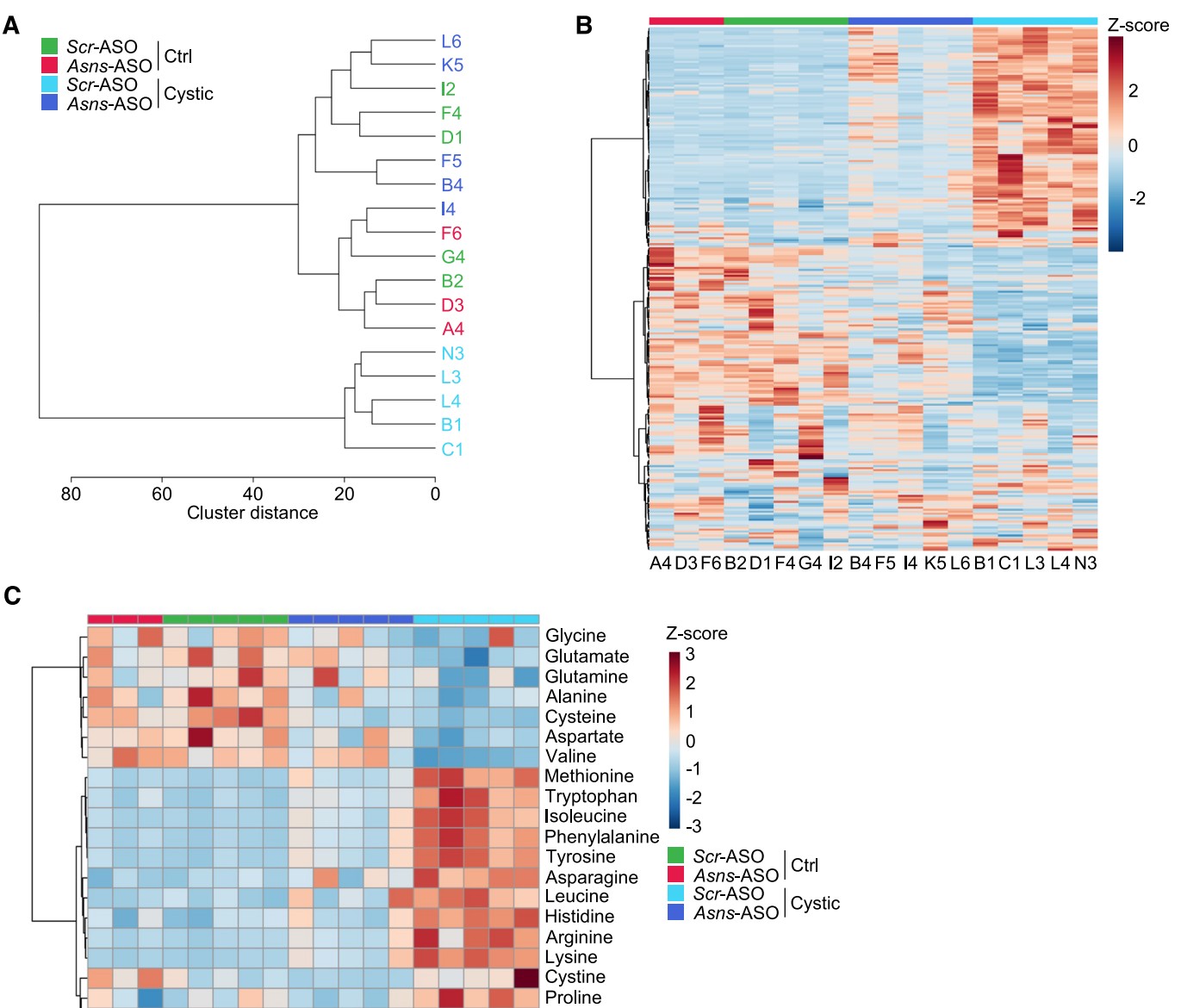

**Figure EV3.  Hierarchical clustering of metabolomics data.**

(A) Dedrogram diagram showing the hierachical clustering of 4 ASO-treated groups of samples analyzed through LC–MS. (B) Heatmap based on the HCA of the metabolom (265 metabolites) comparing *Scr*- and *Asns*-ASO-treated cystic and control kidneys. (C) Heatmap based on the HCA of amino acids detected in PKD cystic and control kidneys treated with *Scr*-ASO or *Asns*-ASO. Data information: in (A–C) clustering based on t test/ANOVA result performed with Metaboanalyst 5.0.

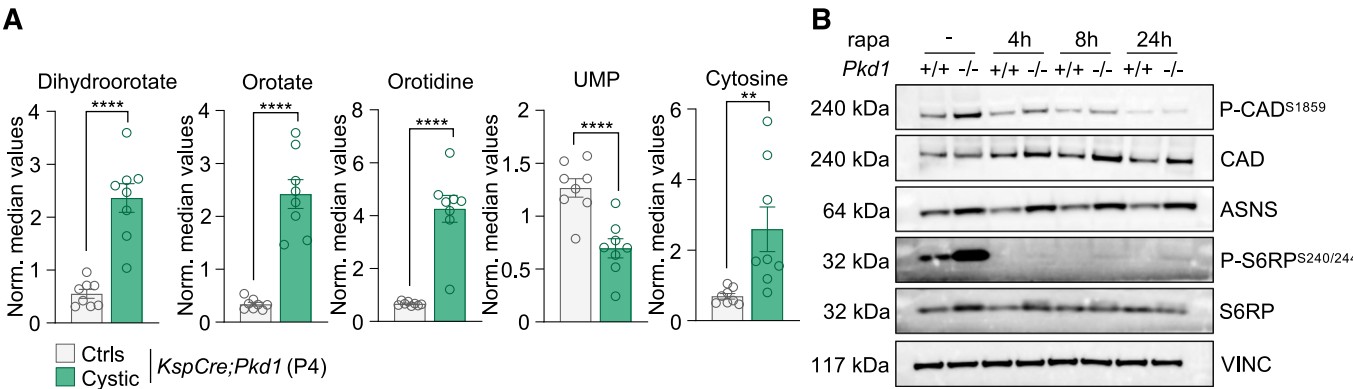

**Figure EV4.** **Validation of the CAD-pyrimidine biosynthesis pathway in different models.**

(A) Intermediate metabolites of de novo pyrimidine biosynthesis pathway analyzed through untargeted metabolomics of *KspCre;Pkd1*$^{\Delta C/flox}$ cystic kidneys and relative controls at P4 ($n = 8$ ctrls; $n = 8$ cystic). (B) P-CAD, ASNS and P-S6RP protein expression in *Pkd1*$^{-/-}$ and control MEF cells treated for 4 h, 8 h, or 24 h with rapamycin (50 nM), after overnight serum starvation ($n = 3$). Data information: in (A) data are shown as mean ± SEM. Student's *t* test. **$P < 0.01$; ****$P < 0.0001$.

