## [Peer Review File · EMBO Molecular Medicine]

Inhibition of Asparagine Synthetase Effectively Retards Polycystic Kidney Disease Progression

Sara Clerici, Christine Podrini, Davide Stefanoni, Gianfranco Distefano, Laura Cassina, Maria Elena Steidl, Laura Tronci, Tamara Canu, Marco Chiaravalli, Daniel Spies, Thomas Bell 3rd, Ana Costa, Antonio Esposito, Angelo D'Alessandro, Christian Frezza, Angela Bachi, and Alessandra Boletta

Corresponding author: Alessandra Boletta (boletta.alessandra@hsr.it)

Review Timeline:

Submission Date:	6th Oct 23
Editorial Decision:	15th Nov 23
Revision Received:	1st Mar 24
Editorial Decision:	5th Apr 24
Revision Received:	11th Apr 24
Accepted:	12th Apr 24

Editor: Zeljko Durdevic

Transaction Report:

15th Nov 2023

Dear Dr. Boletta,

Thank you for the submission of your manuscript to EMBO Molecular Medicine, and please accept my apologies for the delay in getting back to you, which is due to the fact that one referee needed more time to complete his/her review. We have now received feedback from the three reviewers who agreed to evaluate your manuscript. All three referees recognize potential interest of the study but also raise serious criticism that should be addressed in a major revision. If you would like to discuss further the points raised by the referees, I am available to do so via email or video. Let me know if you are interested in this option.

We would welcome the submission of a revised version within three months for further consideration. Please let us know if you require longer to complete the revision.

Please use this link to login to the manuscript system and submit your revision: <https://embomolmed.msubmit.net/cgi-bin/main.plex>

I look forward to receiving your revised manuscript.

Yours sincerely,

Zeljko Durdevic

We require:

- 1) A .docx formatted version of the manuscript text (including legends for main figures, EV figures and tables). Please make sure that the changes are highlighted to be clearly visible.
- 2) Individual production quality figure files as .eps, .tif, .jpg (one file per figure). For guidance, download the 'Figure Guide PDF': (<https://www.embopress.org/page/journal/17574684/authorguide#figureformat>).
- 3) A .docx formatted letter INCLUDING the reviewers' reports and your detailed point-by-point responses to their comments. As part of the EMBO Press transparent editorial process, the point-by-point response is part of the Review Process File (RPF), which will be published alongside your paper.
- 4) A complete author checklist, which you can download from our author guidelines (<https://www.embopress.org/page/journal/17574684/authorguide#submissionofrevisions>). Please insert information in the checklist that is also reflected in the manuscript. The completed author checklist will also be part of the RPF.
- 5) Please note that all corresponding authors are required to supply an ORCID ID for their name upon submission of a revised manuscript.

6) It is mandatory to include a 'Data Availability' section after the Materials and Methods. Before submitting your revision, primary datasets produced in this study need to be deposited in an appropriate public database, and the accession numbers and database listed under 'Data Availability'. Please remember to provide a reviewer password if the datasets are not yet public (see <https://www.embopress.org/page/journal/17574684/authorguide#dataavailability>).

13) Author contributions: You will be asked to provide CRediT (Contributor Role Taxonomy) terms in the submission system. These replace a narrative author contribution section in the manuscript.

14) A Conflict of Interest statement should be provided in the main text.

15) Every published paper now includes a 'Synopsis' to further enhance discoverability. Synopses are displayed on the journal webpage and are freely accessible to all readers. They include a short stand first (maximum of 300 characters, including space) as well as 2-5 one-sentence bullet points that summarize the paper. Please write the bullet points to summarize the key NEW findings. They should be designed to be complementary to the abstract - i.e. not repeat the same text. We encourage inclusion of key acronyms and quantitative information (maximum of 30 words / bullet point). Please use the passive voice. Please attach these in a separate file or send them by email, we will incorporate them accordingly.

Please also suggest a striking image or visual abstract to illustrate your article as a PNG file 550 px wide x 300-800 px high.

**** Reviewer's comments ****

Referee #1 (Remarks for Author):

The manuscript identifies asparagine synthetase (ASNS) as a potential target for ADPKD. Administration of antisense oligonucleotides (ASO) targeting ASNS ameliorated cystic expansion and improved renal function in a late onset ADPKD mouse model. Mechanistic studies show that amino acid deprivation leads to induced transcription of the *Asns* gene in *Pkd1*-null kidneys. Increased expression of ASNS supports cell proliferation via the increased production of glutamine-dependent de novo pyrimidine synthesis. Combination therapy using 2DG and ASO directed *Asns* had a significant effect on suppressing cystic disease. The manuscript has several strengths: 1) demonstration that inhibition of ASNS alone or in combination with inhibition of glycolysis can form a new therapeutic approach to ADPKD. 2) Pharmacologic data are supported by mechanistic data, 3) Metabolic changes and identification of affected pathways are determined using unbiased approaches. And 4) several previous studies from the Boletta lab and others support the use of ASO-*Asns* for the treatment of ADPKD. There are two major weaknesses that can be addressed in a revised manuscript: 1) lack of demonstration that suggested pathway responsible for ASNS upregulation occur in cystic cells, 2) lack of convincing evidence that reduced proliferation is the root cause for the suppression of cystic growth.

Specific points

- 1) Activation of the GCN2-dependent activation of ATF4 needs to be shown in single cells in vivo. It would be important to show that all changes (high levels of p-GCN2, ASNS, ATF4, Ki67-positive cells) occur in cystic cells. Experiments in kidney lysates are definitely informative and valuable, but single cell resolution of the suggested changes would strengthen conclusions.
- 2) Single *Asns*-ASO treatment did not produce a significant drop in kidney volume until P130 (Fig. 2E), but the number of Ki67-positive cells significantly reduced at P94 (Fig. 5B). If reduction of proliferation is the cause of cystic growth retardation, how can these two results be reconciled? Perhaps, the authors need to perform cell apoptosis assays to test whether cell death correlates better with the suppression of cystic growth, unless it is expected that ASNS deprivation drives cells to quiescence. If Ki67-positive cells die off after treatment, why would only *Pkd1*-null cells die? Additional experimentation and discussion would be needed to explain the effect of the treatment on the suppression of cystic expansion.
- 3) A disadvantage of using ASO technology is that it is unknown which cells are targeted, unless fluorescently labeled ASOs are used, which was not the case here. Is there a way that the authors can show that cystic cells are targeted? Can they use RNAscope, immunofluorescence, or something similar to show that ASNS has been successfully targeted in cystic cells? In this regard, it would be important to design the study in a way that treatment is initiated after some obvious symptoms. The authors do acknowledge this as a limitation of the study, but it'd be important to know how reduction of *Asns* mRNA mediates its effects on cystic growth. Does it block de novo cell proliferation or blocks ongoing proliferation driving cells eventually to cell death?
- 4) Fig. 2G. Can the authors double-check stats?
- 5) Fig. 3B and C. Can the authors provide earlier points to see if these changes precede cystic changes.

Referee #2 (Remarks for Author):

In this study the authors build on their prior work highlighting the critical role of glutamine metabolism in the progression of polycystic kidney disease and suggest targeting ASNS, a glutamine-utilizing enzyme in this disease, as a therapeutic target. The authors show that ASNS expression is upregulated in *Pkd1*-deficient MEFs, murine models of PKD and human PKD samples. They present data that this upregulation is the result of activation of the amino acid response branch of the ISR, and that ASNS overexpression is dependent on GCN2, as its silencing results in a reduction in ATF4 and ASNS expression. They convincingly show that silencing ASNS using antisense oligonucleotides mitigates PKD in a mouse PKD model, with reduction of kidney

volume and improved renal function. They suggest that the critical role of ASNS in cyst progression is upregulation of de novo pyrimidine synthesis in support of cellular proliferation. Lastly, they show that combining ASNS ASO with the glycolysis inhibitor 2-DG provides an additive therapeutic effect on PKD.

The key finding in the paper is that systemic delivery of an anti-ASNS ASO mitigates PKD in mice. This is an important finding as it positions ASNS as a therapeutic target, potentially actionable via nucleic acid therapy. A few aspects of the paper should be clarified or strengthened:

- 1) The model suggests that a primary contributor to cell proliferation is upregulation of pyrimidine biosynthesis. However, the metabolomics data show an arguably more prominent downregulation of purine metabolism, suggesting a de-coupling, or perhaps a differential need for these nucleotide pools to sustain proliferation. Can the authors explain or at least speculate why proliferation appears more dependent on pyrimidine metabolism in this context? Similarly, can they show metabolomics data for purine nucleotides and intermediates as they do in Figure 5 for pyrimidines?
- 2) The tracing experiment in Fig 6G showing labeling in carbamoylaspartate from ^{13}C -glutamine is helpful, but does not assess the entire pyrimidine pathway. Labeling in bona fide pyrimidines, including orotate and UMP, should also be shown, and kinetic labeling (i.e. sampling multiple time points) might help assess pathway activity better than a single time point. A labeling experiment using ^{15}N -glutamine would be particularly useful as it would allow concomitant assessment of the purine pathway.
- 3) This also begs the question, if the upregulation of de novo pyrimidine synthesis is a main driver of the PKD phenotype, could a direct inhibitor of this pathway, such as a DHODHi, be a more direct or potentially more potent abrogator of the disease progression? Perhaps the authors could at least speculate about this, because these inhibitors are already available for clinical use.
- 4) The mechanism linking ASNS to enhanced pyrimidine synthesis is not clear. The data are most consistent with ASNS determining aspartate availability, because levels are low in ASNS-expressing cysts and rescued by the ASO. But aspartate is a substrate for pyrimidine synthesis, so its depletion in cystic, proliferative kidney cells seems counterintuitive. The cysts also contain high levels of p-CAD, and this is at least partially reversed by the ASO (Fig. 6A). Is it possible that activation of CAD, an mTORC1 target, is the true mechanistic link?
- 5) At least in the Discussion, it would be helpful to compare/contrast the various routes of glutamine catabolism in PKD with other glutamine-avid tissues. The data seem to indicate that ASNS is a key mechanism by which these cells produce glutamate from glutamine. But most other tissues use GLS for this purpose. Do the authors think that ASNS activity reduces the need for GLS? It would be interesting and surprising if ASNS is really the main mechanism of glutamine catabolism in this setting, as kidneys also express GLS.

Minor:

- 1) In panel 6E, unless I am mistaken, the label should be CAD rather than ASNS.

Referee #3 (Comments on Novelty/Model System for Author):

Most of the work presented in this manuscript represents only incremental work from previously published. The in vitro work is from MEFs, which are not the most representative of tubular cell metabolic needs. The animals used for the in vivo work seem to have a wide phenotype and are not very homogeneously distributed.

Referee #3 (Remarks for Author):

The manuscript by Podrini et al. describes the effect of targeting ASNS as a novel target for therapy acting on a metabolic vulnerability of ADPKD.

In this study, the investigators used antisense oligonucleotides in a slowly progressive orthologous model of PKD to target *Asns* and show a remarkable disease improvement. The investigators identified general control nonderepressible 2 (GCN2) as a major regulator of ASNS increased expression in PKD, and they propose this is mediated by an activation of the amino acid response (AAR) branch of the integrated stress response (ISR). Interestingly, while ASO administration rescued several metabolic pathways, it did not affect glycolysis. Furthermore, the combination of *Asns* inhibition with 2DG further delayed disease progression. Overall, this interesting study shows that targeting ASNS may represent a new and effective therapeutic approach for ADPKD. However, the authors should consider the following comments:

1. It is not clear why the authors use immortalized MEFs for their in vitro studies rather than primary cells from their KO models or iMCD cells or other tubular cells.
2. In vitro studies should provide additional descriptions in their methodologies, such as number of passages at which the experiments were performed, degree of confluency, etc. All these parameters have been shown to alter cell metabolism and should be carefully controlled and noted.
3. It needs to be clarified what the rationale for excluding animals that presented increased kidney volume or presence of cysts

in kidneys and/or in the liver during the first MRI scan, or how many needed to be excluded. This is not clear given that some of the images show cysts at P40, which was when the treatment was initiated and the first MRI.

4. Provide the number and sex of animals included in each experiment.

5. Were there any sex differences?

6. Limitations should be added at the end of the discussion

Please find below our responses to the comments of the reviewers. In black the comments of the reviewers and in blue our responses. We are happy to report a revision that includes comments to all raised issues. To facilitate the re-review of the manuscript we highlight in red text the changes introduced.

***** Reviewer's comments *****

Referee #1 (Remarks for Author):

The manuscript identifies asparagine synthetase (ASNS) as a potential target for ADPKD. Administration of antisense oligonucleotides (ASO) targeting ASNS ameliorated cystic expansion and improved renal function in a late onset ADPKD mouse model. Mechanistic studies show that amino acid deprivation leads to induced transcription of the *Asns* gene in *Pkd1*-null kidneys. Increased expression of ASNS supports cell proliferation via the increased production of glutamine-dependent de novo pyrimidine synthesis. Combination therapy using 2DG and ASO directed *Asns* had a significant effect on suppressing cystic disease. The manuscript has several strengths: 1) demonstration that inhibition of ASNS alone or in combination with inhibition of glycolysis can form a new therapeutic approach to ADPKD. 2) Pharmacologic data are supported by mechanistic data, 3) Metabolic changes and identification of affected pathways are determined using unbiased approaches. And 4) several previous studies from the Boletta lab and others support the use of ASO-*Asns* for the treatment of ADPKD. There are two major weaknesses that can be addressed in a revised manuscript: 1) lack of demonstration that suggested pathway responsible for ASNS upregulation occur in cystic cells, 2) lack of convincing evidence that reduced proliferation is the root cause for the suppression of cystic growth.

We thank the reviewer for the overall positive evaluation of our work. In the revised manuscript we have addressed both major points, as listed below.

Specific points

1) Activation of the GCN2-dependent activation of ATF4 needs to be shown in single cells in vivo. It would be important to show that all changes (high levels of p-GCN2, ASNS, ATF4, Ki67-positive cells) occur in cystic cells. Experiments in kidney lysates are definitely informative and valuable, but single cell resolution of the suggested changes would strengthen conclusions.

Single cell resolution for changes that do not occur at the transcript level (ATF4 and pGCN2) are unfortunately quite difficult to validate at the single cell level. To address this point of the reviewer however, we have screened a series of anti-ASNS antibodies for their good quality in immunohistochemistry. We have identified one anti-ASNS antibody (Cell Signaling Technologies) that responds to the following criteria: i) could stain cells by IF in vitro and the signal was completely nullified by silencing of *Asns* (now shown in Fig. EV1B); ii) could stain kidney tissue from PKD mice and is very significantly reduced upon *Asns*-ASOs. Using this antibody, we are now glad to report that the increased levels of transcript and protein levels of ASNS that we had reported by bulk analysis of whole kidneys corresponds to a clear and strong upregulation of the protein in the cyst-lining epithelia, with minimal staining observed in the interstitium. We now show in two different animal models (*KspCrePkd1^{ΔC/flox}* at P4 and *Tam-Cre;Pkd1^{ΔC/flox}* at P160) that ASNS is strongly upregulated in the cystic epithelia. We conclude that the pathway is upregulated in the cystic epithelia.

2) Single *Asns*-ASO treatment did not produce a significant drop in kidney volume until P130 (Fig. 2E), but the number of Ki67-positive cells significantly reduced at P94 (Fig. 5B). If reduction of proliferation is the cause of cystic growth retardation, how can these two results be reconciled? Perhaps, the authors need to perform cell apoptosis assays to test whether cell death correlates

better with the suppression of cystic growth, unless it is expected that ASNS deprivation drives cells to quiescence. If Ki67-positive cells die off after treatment, why would only *Pkd1*-null cells die? Additional experimentation and discussion would be needed to explain the effect of the treatment on the suppression of cystic expansion.

One should always keep in mind that when measuring proliferation by Ki67 we are taking a snapshot at one precise timepoint of a process that is slowly progressive. No single staining of proliferation can indeed justify the cyst expansion (or reversion) that could be observed in PKD. A minimal increase of proliferation over time would result in cyst expansion (and does result in cyst expansion as demonstrated by multiple groups) even if the proliferation rates observed in PKD are not minimally comparable to the ones observed in cancer, where growth is much more prominent. As an example, the rate of proliferation in the cystic epithelia at P94 is of 10% and it is decreased to 5% in the *Asns*-treated animals. Whereas the increase in kidney weight over controls oscillates between 200% and 400% (Figure EV2D). This is explained by the fact that a slow, but constant increase in proliferation is driving the increase in mass. At each timepoint, however, the rate of proliferation is not massive.

That said, the reviewer is comparing two different animal models. The P94 pilot study was performed upon induction of *Pkd1* inactivation at P24 and analysis of a small number of samples at P94. The P100 MRI capture and volume calculation in the *Asns*-ASOs study shown in the main figures is performed on an animal model in which *Pkd1* inactivation is performed at P45 and the animals sacrificed at P160. This animal model has a much slower progression of disease. This explains, in our view, the fact that at P100 there is only a trend of decrease in volume in the *Asns*-ASOs which is not yet significant in the single treated *Asns*-ASOs it does become strongly significant at P130. We also have measured the rates of apoptosis and we could only detect minimal apoptosis in the PKD kidneys both before and after *Asns*-ASOs (<<1%). Thus, we conclude that the reduced proliferation is likely the mechanism of improved disease progression in PKD animal models upon *Asn*-ASO treatment, but now discuss it more cautiously in view of the fact that, like in any other study on PKD, we could not find a linear correlation between the rates of proliferation and the disease progression or improvement.

3) A disadvantage of using ASO technology is that it is unknown which cells are targeted, unless fluorescently labeled ASOs are used, which was not the case here. Is there a way that the authors can show that cystic cells are targeted? Can they use RNAscope, immunofluorescence, or something similar to show that ASNS has been successfully targeted in cystic cells? In this regard, it would be important to design the study in a way that treatment is initiated after some obvious symptoms. The authors do acknowledge this as a limitation of the study, but it'd be important to know how reduction of *Asns* mRNA mediates its effects on cystic growth. Does it block de novo cell proliferation or blocks ongoing proliferation driving cells eventually to cell death?

Thanks for the suggestion. We have used staining of the anti-sense oligonucleotides using an antibody developed by IONIS specifically against the chemical moiety carried by their proprietary ASOs, and we now provide evidence that the ASOs can penetrate in multiple different regions of the kidney, including the cystic epithelia, although this distribution is non homogeneous, in line with the known biodistribution of ASOs in the kidney and explaining why rescue of the cystic tissue is also non-homogeneous in our treatment. The ASOs do reach the cystic epithelia though. We include these data in Fig. EV2G.

4) Fig. 2G. Can the authors double-check stats?

Statistical analysis was correct. The outlier is causing this lack of significance. Removing it would indeed make the difference significant and further strengthen our findings. But we prefer to leave all raw data included into the graphs. There is a trend, which is quite evident. And the BUN is stringly significant. Thanks for noticing this.

5) Fig. 3B and C. Can the authors provide earlier points to see if these changes precede cystic changes.

We cannot conclude from our data whether the activation of this pathway precedes or follows the formation of cysts. Therefore, we cannot draw any conclusion on this specific aspect at the moment. However, we do observe minimally dilated tubules in the P4 kidneys that are already strongly positive to ASNS staining (Figure 2D). We do not think this is sufficient to draw conclusive evidence and we will need much more robust analysis using single cell RNA seq studies currently being developed in the lab. It is important to note, however, that to date there is no single pathway reported to precede cyst formation in *Pkd1* mutant cells. Thus, we believe that these studies will resolve an important question mark in the field and we are working on that. This does not take away the fact that targeting ASNS is a good strategy to design a therapy for PKD.

Referee #2 (Remarks for Author):

In this study the authors build on their prior work highlighting the critical role of glutamine metabolism in the progression of polycystic kidney disease and suggest targeting ASNS, a glutamine-utilizing enzyme in this disease, as a therapeutic target. The authors show that ASNS expression is upregulated in *Pkd1*-deficient MEFs, murine models of PKD and human PKD samples. They present data that this upregulation is the result of activation of the amino acid response branch of the ISR, and that ASNS overexpression is dependent on GCN2, as its silencing results in a reduction in ATF4 and ASNS expression. They convincingly show that silencing ASNS using antisense oligonucleotides mitigates PKD in a mouse PKD model, with reduction of kidney volume and improved renal function. They suggest that the critical role of ASNS in cyst progression is upregulation of de novo pyrimidine synthesis in support of cellular proliferation. Lastly, they show that combining ASNS ASO with the glycolysis inhibitor 2-DG provides an additive therapeutic effect on PKD.

The key finding in the paper is that systemic delivery of an anti-ASNS ASO mitigates PKD in mice. This is an important finding as it positions ASNS as a therapeutic target, potentially actionable via nucleic acid therapy. A few aspects of the paper should be clarified or strengthened:

1) The model suggests that a primary contributor to cell proliferation is upregulation of pyrimidine biosynthesis. However, the metabolomics data show an arguably more prominent downregulation of purine metabolism, suggesting a de-coupling, or perhaps a differential need for these nucleotide pools to sustain proliferation. Can the authors explain or at least speculate why proliferation appears more dependent on pyrimidine metabolism in this context? Similarly, can they show metabolomics data for purine nucleotides and intermediates as they do in Figure 5 for pyrimidines?

We agree with this reviewer. We cannot certainly say that the metabolic alterations corrected by silencing ASNS are confined to the pyrimidine biosynthesis pathway. We do observe a bunch of different metabolic alterations. The link between ASNS upregulation and increased proliferation has been previously attributed to the pyrimidine biosynthesis pathway and here we simply wanted to show that a similar effect as the one previously reported is visible in PKD. We now make it more clear and avoid over-stating the potential role of this change in PKD. In addition, we include data on the purine pathway as suggested by the reviewer in Figure 5C.

2) The tracing experiment in Fig 6G showing labeling in carbamoylaspartate from ¹³C-glutamine is helpful, but does not assess the entire pyrimidine pathway. Labeling in bona fide pyrimidines, including orotate and UMP, should also be shown, and kinetic labeling (i.e. sampling multiple time points) might help assess pathway activity better than a single time point. A labeling experiment using ¹⁵N-glutamine would be particularly useful as it would allow concomitant assessment of the purine pathway.

Thanks for the suggestion, we now include tracing using ^{15}N -glutamine. The data indeed support the role of Asns in fueling nitrogen to the pyrimidine biosynthesis pathway (data introduced in Fig 6G) and also show the increase in downstream pyrimidine intermediates such as orotate and UMP. Of interest, we have performed silencing of Asns and this demonstrates that upon silencing the contribution of the glutamine nitrogen group to the pyrimidine biosynthesis pathway is reduced.

3) This also begs the question, if the upregulation of de novo pyrimidine synthesis is a main driver of the PKD phenotype, could a direct inhibitor of this pathway, such as a DHODHi, be a more direct or potentially more potent abrogator of the disease progression? Perhaps the authors could at least speculate about this, because these inhibitors are already available for clinical use.

This is an excellent point. As stated above, we consider it very unlikely that the increased pyrimidine biosynthesis is the sole pathway responsible for the increased proliferation. It might be one of the pathways involved. However, treatment with the indicated inhibitor did impact very severely on *Pkd1* cells *in vitro*, but in our hands also on the control cells (see figure below). Nevertheless, the reviewer is correct and we have introduced a sentence to suggest that DHODH inhibitors might provide some benefit in PKD and could be tested *in vivo* in the future.

Figure for reviewers removed

4) The mechanism linking ASNS to enhanced pyrimidine synthesis is not clear. The data are most consistent with ASNS determining aspartate availability, because levels are low in ASNS-expressing cysts and rescued by the ASO. But aspartate is a substrate for pyrimidine synthesis, so its depletion in cystic, proliferative kidney cells seems counterintuitive. The cysts also contain high levels of p-CAD, and this is at least partially reversed by the ASO (Fig. 6A). Is it possible that activation of CAD, an mTORC1 target, is the true mechanistic link?

We now include the tracing with the $^{15}\text{N}_2$ -glutamine to show that indeed the CAD activation results in enhanced utilization of glutamine-derived nitrogen for pyrimidine biosynthesis. Also, the reviewer has an excellent point here as mTORC1 is upregulated in PKD cells and mTORC1 is known to activate CAD. Indeed, when we treated *Pkd1* mutant cells with rapamycin, this treatment was able to reduce the activation levels of pCAD (now shown in Fig. EV4B). Of great interest, rapamycin did not rescue the upregulation of Asns in these cells. These data are extremely interesting as they suggest that the two pathways cooperate to enhance pyrimidine biosynthesis in PKD, but they are not interdependent.

For what concerns the reduced levels of aspartate in the PKD kidneys, we think this might provide evidence that there is an increased utilization of aspartate to produce asparagine. In our view, observing reduced levels of aspartate in a snapshot metabolomics does not necessarily mean that there is reduced availability, but rather that there is an increased utilization, because aspartate is utilized to produce asparagine. And we believe that the fact that aspartate levels are rescued upon silencing of Asns (and so does the increased asparagine) supports our model.

5) At least in the Discussion, it would be helpful to compare/contrast the various routes of glutamine

catabolism in PKD with other glutamine-avid tissues. The data seem to indicate that ASNS is a key mechanism by which these cells produce glutamate from glutamine. But most other tissues use GLS for this purpose. Do the authors think that ASNS activity reduces the need for GLS? It would be interesting and surprising if ASNS is really the main mechanism of glutamine catabolism in this setting, as kidneys also express GLS.

Curiously, GLS seems to be less relevant in PKD because it is not consistently upregulated in all PKD datasets analyzed (see panel below), as opposed to ASNS which is upregulated in all possible settings as we previously published in Podrini et al, 2018 and as shown in Figure 1. This is interesting also in light of the fact that two previous studies had tried to treat PKD mice with the GLS inhibitor CB-839 reporting inconsistent results in different animal models (Flowers et al, and Soomro et al, see references). Also, we have shown that silencing *Asns* completely impairs the utilization of glutamine in the TCA cycle and the conversion to a-KG (Podrini et al, 2018). We show now in figure 6 that tracing using the ¹⁵N₂-glutamine confirms that silencing of *Asns* completely rescues the increased utilization of glutamine by PKD cells. Nevertheless, we agree with the reviewer that a careful and parallel analysis of the ASNS vs GLS contribution in PKD should be performed and we plan to do so in the future. However, we would also like to point out here that there is a precedent for a condition in which ASNS is the main driver of glutamine utilization in cells, and this is the endothelial cells in a neovascularization setting as shown by Dr. Peter Carmeliet (Huang et al, EMBO Journal, 2017; doi: [10.15252/emj.201695518](https://doi.org/10.15252/emj.201695518)). We think that PKD cells similarly rely on ASNS for glutamine utilization.

Figure for reviewers removed

Minor:

1) In panel 6E, unless I am mistaken, the label should be CAD rather than ASNS.
We have corrected, sorry for the confusion.

Referee #3 (Comments on Novelty/Model System for Author):

Most of the work presented in this manuscript represents only incremental work from previously published.

We respectfully disagree. There was no evidence prior to the current work that inhibiting ASNS could be beneficial in the setting of PKD. While we had previously identified this enzyme as upregulated in cells, it was not necessarily obvious that its silencing *in vivo* would be beneficial for PKD progression. And certainly not to this extent.

The *in vitro* work is from MEFs, which are not the most representative of tubular cell metabolic needs.

We added data on renal epithelial cells, see below.

The animals used for the *in vivo* work seem to have a wide phenotype and are not very homogeneously distributed.

The PKD models tend to be quite variable in their phenotype. In a way this is a plus, as patients affected by this disease also manifest with a quite variable phenotype. Nevertheless, the power of the study was calculated and the numerosity utilized clearly shows the improved phenotype.

Referee #3 (Remarks for Author):

The manuscript by Podrini et al. describes the effect of targeting ASNS as a novel target for therapy acting on a metabolic vulnerability of ADPKD.

In this study, the investigators used antisense oligonucleotides in a slowly progressive orthologous model of PKD to target Asns and show a remarkable disease improvement. The investigators identified general control nonderepressible 2 (GCN2) as a major regulator of ASNS increased expression in PKD, and they propose this is mediated by an activation of the amino acid response (AAR) branch of the integrated stress response (ISR). Interestingly, while ASO administration rescued several metabolic pathways, it did not affect glycolysis. Furthermore, the combination of Asns inhibition with 2DG further delayed disease progression. Overall, this interesting study shows that targeting ASNS may represent a new and effective therapeutic approach for ADPKD. However, the authors should consider the following comments:

1. It is not clear why the authors use immortalized MEFs for their *in vitro* studies rather than primary cells from their KO models or iMCD cells or other tubular cells.

In our opinion MEFs are a quite good model for studying *Pkd1* function, as these cells express higher levels of polycystin than the epithelial cells (Wodarczyk et al, PLoS One, 2009; Castelli et al, *Nat Comm*, 2013). Also, we believe that a general function for this gigantic receptor should be conserved in all cellular systems utilized if relevant and central to the core function of the gene studied. The reason why PKD manifests with cysts originating from the epithelia is because these acquire a second hit (somatic mutation) that lowers its function specifically in these cells, but the protein is quite extensively expressed in multiple tissues. In line with this, we and others have shown in the past that fibroblasts recapitulate alterations observed in epithelia. Furthermore, in the current study, we have validated several of our key findings *in vivo*. Nevertheless, we agree with the reviewer that providing evidence that the same pathway and alterations are observed in renal epithelia in cell culture strengthens our conclusions and we now provide data from a newly generated renal epithelial cell line (mCCD cells) in which we have inactivated the *Pkd1* gene by CRISPR/Cas9. Description of these cell lines is included now in the manuscript and we show in figure 1C that they display increased levels of asparagine synthetase. We are currently completing the characterization of these cells (Cassina et al in preparation), as they consistently display the reduced OXPHOS and increased glycolysis that we and others have previously reported in MEFs and other cell types. These studies will be completed in a separate setting. In addition, we also show that the upregulation of ASNS occurs in the epithelia lining the cysts in PKD kidneys (Figure 2D).

2. In vitro studies should provide additional descriptions in their methodologies, such as number of passages at which the experiments were performed, degree of confluency, etc. All these parameters have been shown to alter cell metabolism and should be carefully controlled and noted.

We have added all specifics on cell culture data and on how cells are passaged and cultured.

3. It needs to be clarified what the rationale for excluding animals that presented increased kidney volume or presence of cysts in kidneys and/or in the liver during the first MRI scan, or how many needed to be excluded. This is not clear given that some of the images show cysts at P40, which was when the treatment was initiated and the first MRI.

The tamoxifen-inducible system can be tricky to use. The Cre reporters are all very sensitive to ambient tamoxifen. So, we exclude from the analysis all animals that have a cystic phenotype even before inactivation with tamoxifen has been induced. The reason for doing so is that for the *Pkd1* gene inactivation at different timepoints can induce major differences in phenotype. For animals that present with cysts prior to tamoxifen induction we would not know when the ambient-induced inactivation has occurred, introducing a bias in the experiment. We now explain these aspects in detail in the methods.

4. Provide the number and sex of animals included in each experiment.

We now show in all graphs females and males and provide numbers.

5. Were there any sex differences?

In this animal model, we could not observe significant differences based on animal sex.

6. Limitations should be added at the end of the discussion

We now introduced a paragraph on limitations in the discussion.

5th Apr 2024

Dear Dr. Boletta,

Thank you for the submission of your revised manuscript to EMBO Molecular Medicine. I am pleased to inform you that we will be able to accept your manuscript pending the following final amendments:

- 1) Authors: We note a discrepancy of author's name: Thomas A. Bell in the manuscript and Alex Bell in our submission system. Please correct.
- 2) Figures:
 - Please upload individual, high-resolution files in TIFF, EPS or PDF format for each main and EV figures. For more information on figure presentation please check "Author Guidelines".
<https://www.embopress.org/page/journal/17574684/authorguide#datapresentationformat>
 - Please add information in the legend of Appendix Figure S1 that the images B1, I1, L4, B4, I4 and L6 are also presented in Figure 2L.
- 3) In the main manuscript file, please do the following:
 - Please address all comments suggested by our data editors listed below:
 - o Figure legends:
 1. Please note that the statistical test related information in the legend of figure 6f-g is incorrectly labelled as 6a-d. This needs to be rectified.
 2. Please note that in figure 4c; there is a mismatch between the annotated p values in the figure legend and the annotated p values in the figure file that should be corrected.
 3. Please note that information related to n is missing in the legends of figures 2b, f-g, j-k; 4c; 5a-c; 6f-g; 7g; EV 1c; EV 2b, d-f; EV 4a.
 4. Although 'n' is provided, please describe the nature of entity for 'n' in the legends of figures 3d-e.
 5. Please note that the scale bar needs to be defined for figures 2e; 4a; 7c.
 - Add up to 5 keywords.
 - Remove "unpublished" (p.13).
 - Provide the antibody dilutions that were used for each antibody.
 - Statistical paragraph should reflect all information that you have filled in the Authors Checklist, especially regarding randomization, blinding, replication etc.
 - Rename "Competing interest" to "Disclosure and competing interests statement". We updated our journal's competing interests policy in January 2022 and request authors to consider both actual and perceived competing interests. Please review the policy <https://www.embopress.org/competing-interests> and update your competing interests if necessary. Please add the sentence "Luigi Naldini is a member of the journal's editorial board. This has no bearing on the editorial consideration of this article for publication."
 - Author contributions: Please remove it from the manuscript and specify author contributions in our submission system. CRediT has replaced the traditional author contributions section because it offers a systematic machine-readable author contributions format that allows for more effective research assessment. You are encouraged to use the free text boxes beneath each contributing author's name to add specific details on the author's contribution. More information is available in our guide to authors:
<https://www.embopress.org/page/journal/17574684/authorguide#authorshipguidelines>
 - Please be aware that all deposited datasets should be made freely available upon acceptance, without restriction. Please check "Author Guidelines" for more information.
<https://www.embopress.org/page/journal/17574684/authorguide#availabilityofpublishedmaterial>
- 4) Appendix: Please add title page with the table of content and page numbers.
- 5) Synopsis: Every published paper now includes a 'Synopsis' to further enhance discoverability. Synopses are displayed on the journal webpage and are freely accessible to all readers. They include separate synopsis image and synopsis text.
 - Synopsis image: Please provide a striking image or visual abstract as a high-resolution jpeg file 550 px-wide x (250-400)-px high to illustrate your article.
 - Synopsis text: Please provide a short standfirst (maximum of 300 characters, including space) as well as 2-5 one sentence bullet points that summarise the paper as a .doc file. Please write the bullet points to summarise the key NEW findings. They should be designed to be complementary to the abstract - i.e. not repeat the same text. We encourage inclusion of key acronyms and quantitative information (maximum of 30 words / bullet point). Please use the passive voice.
 - Please check your synopsis text and image before submission with your revised manuscript. Please be aware that in the proof stage minor corrections only are allowed (e.g., typos).
- 6) For more information: This space should be used to list relevant web links for further consultation by our readers. Could you identify some relevant ones and provide such information as well? Some examples are patient associations, relevant databases, OMIM/proteins/genes links, author's websites, etc...
- 7) Source data: Please upload one folder per figure.
- 8) As part of the EMBO Publications transparent editorial process initiative (see our Editorial at <http://embomolmed.embopress.org/content/2/9/329>), EMBO Molecular Medicine will publish online a Review Process File (RPF)

to accompany accepted manuscripts. This file will be published in conjunction with your paper and will include the anonymous referee reports, your point-by-point response and all pertinent correspondence relating to the manuscript. Let us know whether you agree with the publication of the RPF and as here, if you want to remove or not any figures from it prior to publication. Please note that the Authors checklist will be published at the end of the RPF.

9) Please provide a point-by-point letter INCLUDING my comments as well as the reviewer's reports and your detailed responses (as Word file).

I look forward to reading a new revised version of your manuscript as soon as possible.

Yours sincerely,

Zeljko Durdevic

*** Instructions to submit your revised manuscript ***

1) a .docx formatted version of the manuscript text (including Figure legends and tables)

2) Separate figure files*

3) supplemental information as Expanded View and/or Appendix. Please carefully check the authors guidelines for formatting Expanded view and Appendix figures and tables at <https://www.embopress.org/page/journal/17574684/authorguide#expandedview>

4) a letter INCLUDING the reviewer's reports and your detailed responses to their comments (as Word file).

5) The paper explained: EMBO Molecular Medicine articles are accompanied by a summary of the articles to emphasize the major findings in the paper and their medical implications for the non-specialist reader. Please provide a draft summary of your article highlighting

6) For more information: There is space at the end of each article to list relevant web links for further consultation by our readers. Could you identify some relevant ones and provide such information as well? Some examples are patient associations, relevant databases, OMIM/proteins/genes links, author's websites, etc...

7) Author contributions: the contribution of every author must be detailed in a separate section.

8) EMBO Molecular Medicine now requires a complete author checklist (<https://www.embopress.org/page/journal/17574684/authorguide>) to be submitted with all revised manuscripts. Please use the checklist as guideline for the sort of information we need WITHIN the manuscript. The checklist should only be filled with page numbers where the information can be found. This is particularly important for animal reporting, antibody dilutions (missing) and exact values and n that should be indicated instead of a range.

9) Every published paper now includes a 'Synopsis' to further enhance discoverability. Synopses are displayed on the journal webpage and are freely accessible to all readers. They include a short stand first (maximum of 300 characters, including space) as well as 2-5 one sentence bullet points that summarise the paper. Please write the bullet points to summarise the key NEW findings. They should be designed to be complementary to the abstract - i.e. not repeat the same text. We encourage inclusion of key acronyms and quantitative information (maximum of 30 words / bullet point). Please use the passive voice. Please attach these in a separate file or send them by email, we will incorporate them accordingly.

You are also welcome to suggest a striking image or visual abstract to illustrate your article. If you do please provide a jpeg file 550 px-wide x 300-800px high.

10) A Conflict of Interest statement should be provided in the main text

11) Please note that we now mandate that all corresponding authors list an ORCID digital identifier. This takes <90 seconds to complete. We encourage all authors to supply an ORCID identifier, which will be linked to their name for unambiguous name identification.

Currently, our records indicate that the ORCID for your account is 0000-0002-4704-4006.

Link Not Available

Photos 400-800 DPI

*Additional important information regarding figures and illustrations can be found at <https://bit.ly/EMBOPressFigurePreparationGuideline>. See also figure legend preparation guidelines: <https://www.embopress.org/page/journal/17574684/authorguide#figureformat>

***** Reviewer's comments *****

Referee #1 (Remarks for Author):

The authors have satisfactorily addressed my concerns, either by appropriate experimentation or by addressing them in the discussion. I support the publication of the manuscript in EMBO Molecular Medicine.

Referee #2 (Comments on Novelty/Model System for Author):

I find the revised paper to be convincing in terms of the models used and the conclusions drawn.

Referee #2 (Remarks for Author):

The revision addresses my critiques.

The authors addressed the remaining editorial issues.

12th Apr 2024

Dear Dr. Boletta,

We are pleased to inform you that your manuscript is accepted for publication and is now being sent to our publisher to be included in the next available issue of EMBO Molecular Medicine. Please be aware that all deposited datasets should be made freely available upon acceptance.
